# SPATIAL REASONING WITH MLLMS: A NEW PATH TO GRAPH-STRUCTURED OPTIMIZATION

## ABSTRACT

Graph-structured problems pose significant challenges due to their complex structures and large scales, often making traditional computational approaches suboptimal or costly. However, when these problems are visually represented, humans can often solve them more intuitively, leveraging our inherent spatial reasoning capabilities. In this work, we introduce an original and novel approach by feeding graphs as images into multimodal large language models (MLLMs), aiming for a loss-free representation that preserves the graph's structural integrity and enables machines to mimic this human-like thinking. Our pioneering exploration of MLLMs addresses various graph-structured challenges, from combinatorial tasks like influence maximization to sequential decision-making processes such as network dismantling, along with tackling six basic graph-related problems. Our experiments reveal that MLLMs possess remarkable spatial intelligence and a unique aptitude for these problems, marking a significant step forward in enabling machines to understand and analyze graph-structured data with human-like depth and intuition. These findings also suggest that combining MLLMs with straightforward optimization techniques could offer a new, effective paradigm for managing large-scale graph problems without complex derivations, computationally demanding training and fine-tuning.

## 1 INTRODUCTION

Graph-structured problems are crucial across various fields due to their ability to model complex relationships (Lü et al., 2016; Artime et al., 2024; Grassia et al., 2021). In social networks, identifying key nodes can improve information dissemination and marketing strategies (Kempe et al., 2003). Public health also benefits, as targeting influential nodes helps develop effective immunization strategies to prevent disease spread (Chen et al., 2008). Meanwhile, graph-structured problems are challenging because, unlike traditional Euclidean problems that leverage geometric properties for optimization, graphs are discrete structures lacking clear spatial relationships. This irregularity complicates the application of standard continuous optimization methods. In real-world applications, many graph-structured problems are NP-hard. As the number of nodes and edges grows, the combinatorial explosion of possible configurations renders brute-force methods impractical within a reasonable timeframe.

Meta-heuristic algorithms (Gong et al., 2016b; Zhao et al., 2023) are effective for complicated problems but face scalability challenges with large datasets. As the problem size increases, the search space expands exponentially, making it harder to find optimal solutions efficiently. Moreover, evaluating solutions is computationally expensive, especially when many iterations are required, further limiting their scalability. Recent years have witnessed incredible progress in the use of graph neural networks (GNNs) on many graph-related tasks like node classification (Kipf & Welling, 2016; Veličković et al., 2017) and graph classification (Jin et al., 2020; Han et al., 2022). However, GNNs may lose global structural information due to over-smoothing (Chen et al., 2020), where repeated message passing can cause node representations to become indistinguishable, limiting their performance on large-scale networks. In addition, many real-world networks inherently lack labeled data, making it challenging for GNNs to learn meaningful embeddings effectively. Since GNNs are typically trained on specific graph structures, their ability to generalize to unseen networks is limited, further hindering their applicability when applied to various networks. As indicated in a recent study (Angelini & Ricci-Tersenghi, 2023), the performance of modern GNN-based methods is sometimes

even worse than simple greedy algorithms, implying that GNNs may not be the optimal backbone for graph-structured combinatorial problems.

Recently, the emergence of large language models (LLMs) has achieved tremendous improvements in many areas such as sentiment analysis (Deng et al., 2023), translation (Gong et al., 2024), optimization (Romera-Paredes et al., 2024), medical applications (Chervenak et al., 2023) and social science (Zhang et al., 2024), etc. Therefore, it is natural to consider whether the success of LLMs in other fields can be replicated in graph-related tasks (Chen et al., 2024; Tang et al., 2024). As illustrated by (Fatemi et al., 2023; Wang et al., 2024), LLMs are not good at understanding graph-structured data and cannot even deliver acceptable results on some basic tasks. Moreover, LLMs' performance drops drastically with the increase in the graph size. Consequently, it is unlikely that LLMs can directly tackle complex problems in real-world networks at the present stage.

Over time, the representation of graph-structured data has evolved significantly with the development of computational techniques, as illustrated in Figure 1. Initially, (meta)heuristic methods focus on directly manipulating graph data through adjacency matrices. Representation learning progressed significantly, as demonstrated by Graph Neural Networks (GNNs), which utilize low-dimensional vector spaces to capture the structural properties of graphs, enabling more complex computations. In the era of LLMs, the fundamental way of representing graph-structured data shifted to natural language, allowing machines to interpret and analyze graphs through textual descriptions. However, graphs are inherently spatial constructs, where the placement, distance, and connections reveal abundant information about the system's structure. Converting a graph into non-visual formats such as adjacency matrices, texts, or embeddings will obscure and lose some structural details, particularly global and high-order information.

**(a) (Meta)heuristics**    **(b) GNNs**    **(c) LLMs**    **(d) MLLMs**

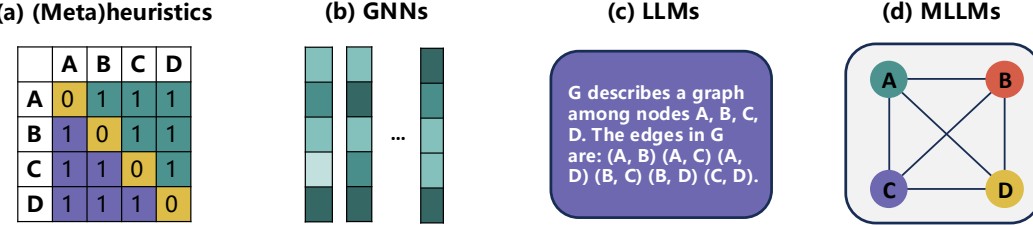

Figure 1: The representation of different eras of graph structure. (a) Adjacency matrix; (b) Embedding; (c) Text; (d) Image.

In fact, certain problems that are highly complex for machines may be far less challenging for humans, a phenomenon particularly evident in combinatorial optimization. When graph data is properly visualized, humans can use our innate spatial and visual reasoning to effectively tackle these problems. As the advent of multimodal large language models (MLLMs), we may stand on the brink of a transformative shift in tackling such complex problems. Images, as low-loss (potentially loss-free with advancements in visualization) representations of graph structures, can now be processed by machines, enabling them to directly comprehend and analyze graph data like humans.

In this study, we strategically utilize MLLMs to address a range of challenges, from sequential decision-making in network dismantling (ND) to complex combinatorial problem influence maximization (IM) to demonstrate their unique strengths in handling graph-structured problems. The results are highly promising with MLLMs exhibiting remarkable spatial intelligence and delivering outstanding performance on these complex tasks, all without the need for fine-tuning, suggesting a new era for dealing graph-structured problems may be approaching. Given their simplicity and effectiveness, MLLMs combined with basic optimization techniques hold great potential as a practical solution for tackling complex graph-structured problems in the future. Furthermore, we explore MLLMs' performance on fundamental graph problems, identifying key factors to their effectiveness. We also discuss potential directions for further unlocking the vast potential of MLLMs in this domain.

In visualization, we tailor the strategies to accommodate different network sizes. The structural information of the tested networks is shown in Table 1. For small networks (less than 150 nodes), we display labels for all nodes in the images provided to the MLLMs, referred to as full-label. For large-scale networks, displaying labels for every node is impractical due to the limited canvas size. In these cases, we selectively label only the nodes most likely to be critical, referred to as partial-label. For the network dismantling problem, we use a simple prompt for the MLLMs and find that it is

sufficient to achieve excellent performance, showing the model's inherent spatial intelligence without requiring complex instructions. For influence maximization, we adopt an agent-modeling framework that directs the MLLMs to select seed nodes with varying biases. Our experimental results with parameter setting, and full details of four method are given in Section A (Appendix).

Table 1: The structural information of tested real-world networks after removing self-loops and isolated components. $|\mathcal{V}|$ and $|\mathcal{E}|$ refer to the number of nodes and edges, respectively.

| Network | Karate | Dolphins | Lesmis | Polbooks | Facebook | Router | Sex |
|---|---|---|---|---|---|---|---|
| $|\mathcal{V}|$ | 34 | 62 | 77 | 105 | 4,039 | 5,022 | 15,810 |
| $|\mathcal{E}|$ | 78 | 159 | 254 | 441 | 88,234 | 6,258 | 35,840 |

## 2 NETWORK DISMANTLING

**Network Dismantling (ND)** ais to identify a minimal set of nodes $S \subset V$ whose removal causes a significant reduction in the size of the largest connected component, effectively fragmenting the network. Given a network with $N$ nodes, the robustness defined as: $R = \frac{1}{N} \sum_{Q=1}^{N} s(Q)$, where $s(Q)$ represents the size of the largest connected component after the removal of $Q$ nodes.

**MLLMs possess a strong grasp of graph structure:** Figure 2 illustrates an attempt of the network dismantling process guided by an MLLM. In traditional approaches like degree centrality, the nodes with the highest degree, such as 32 or 33, would be prioritized for removal to minimize the size of the largest connected component (LCC). However, the MLLM suggests removing node 0 first, which leads to a more rapid reduction in the LCC size, immediately to 27. This result implies the MLLM's ability to predict the cascading effects of node removal beyond the most intuitive observation (degree).

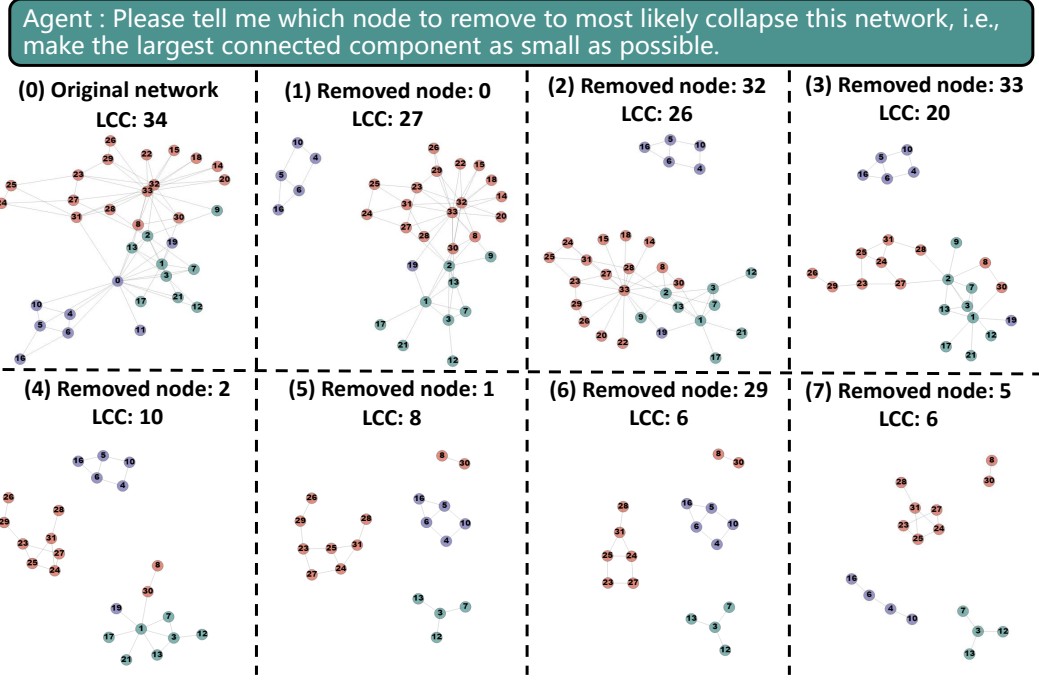

Figure 2: The diagram of network dismantling guided by MLLM on the Karate network. The network is iteratively fed into the MLLM as an image to obtain suggestions for the next node to remove. The layout will dynamically adjust in response to changes in the network structure.

**Network size will affect the decision robustness of MLLMs:** In the Karate network, the MLLMs show a relatively concentrated pattern of node removal, reflected by the dark color of the diagonal elements in Figure 3. The growing size and complexity of networks likely hinder the MLLMs' ability to pinpoint a single set of critical nodes such as Polbooks. The differing removal frequencies suggest that the MLLMs' selections will be more varied, likely due to the difficulties in visual identification.

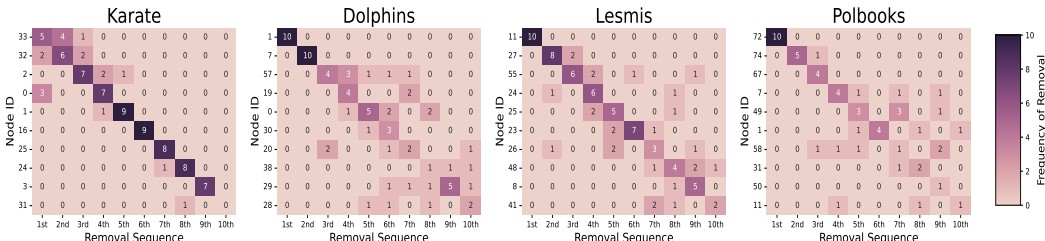

Figure 3: The frequency of node removal using MLLMs for network dismantling. Each cell shows the frequency with which each node (y-axis) was removed at a particular sequence position (x-axis) over ten tests.

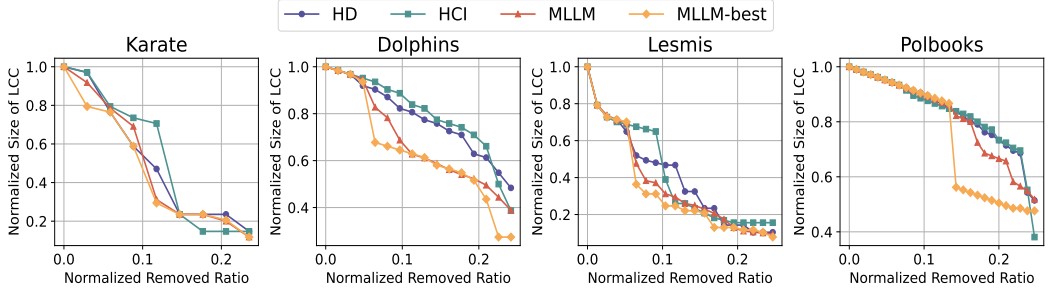

Figure 4: The comparative performance on the normalized size of the Largest Connected Component (LCC) of four methods in network dismantling. MLLM refers to the average performance over ten attempts using Multi-Modal Large Language Models and MLLM-best is the best result among ten attempts. The dismantling process stops after 25% nodes are removed.

**MLLMs can beat traditional methods with its inherent intelligence:** Figure 4 presents the results of network dismantling on different networks. Note that the MLLMs are currently only applicable to the full-label case due to the lack of interactive channels between MLLMs and the visualization tools. The results demonstrate that both the MLLM and MLLM-best consistently outperform traditional methods such as HD and HCI in reducing the LCC size.

Table 2: The area under the curve (AUC) of different node removal strategies across networks.

| Network | Karate | Dolphins | Lesmis | Polbooks |
|---|---|---|---|---|
| Degree | 4.07 | 11.77 | 7.62 | 21.85 |
| CI | 4.31 | 12.13 | 7.80 | 21.81 |
| MLLM | 3.94 | 10.28 | 6.88 | 21.27 |
| MLLM-best | **3.67** | **9.67** | **6.33** | **19.41** |

Table 4 presents the AUC for the normalized size of the LCC (in Figure 4) with lower AUC values indicating better result. Not only the MLLM-best but also MLLM consistently shows the lowest AUC across networks, demonstrating its effectiveness in network dismantling.

## 3 INFLUENCE MAXIMIZATION

**Influence Maximization (IM)** aims to find a subset of seed nodes $S \subset V$ that maximizes the overall influence spread across a network. This spread is governed by a probabilistic diffusion model. The goal of the problem can be formally expressed as: Maximize $\sigma(S)$, where $\sigma(S)$ denotes the expected spread of influence starting from the seed set $S$.

### 3.1 SMALL-SCALE NETWORK

In this section, we employ an agent-based method for IM. Each agent is equipped with unique criteria. The visualization method and agent vary with network sizes. Unlike the ND task, where nodes are selected sequentially, seed nodes in IM are selected simultaneously, introducing additional challenges: (1) MLLMs must account for the global pattern and interconnections among seeds; (2) The selected seeds must satisfy specific requirements, such as seed size, and ensure no repetition.

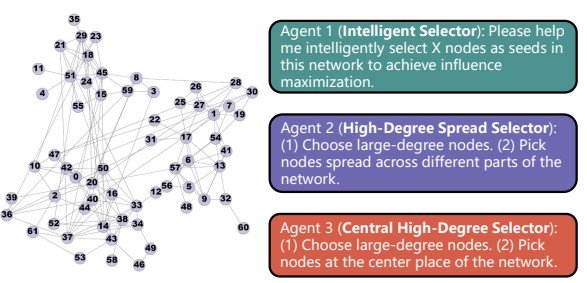

Figure 5 shows the MLLM-based IM in small-scale networks where all nodes are visualized on a single canvas with labeled node IDs. The full-label network will be input to MLLM as an image for multiple-node selection in one go. We design each agent focusing on a different criterion. Agent 1 solely relies on the intelligence of MLLM while Agents 2 and 3 are equipped with specific hints, focusing on the distributed and central parts, respectively. The prompt for Agent 1

Figure 5: The illustrations of four agents for IM on small-scale networks. The full-label network (left) will be inputted into MLLM along with the prompts for agents (right).

is also placed in front of the prompt for the other agents as the leading sentence to explain the task.

Table 3: The validations across different networks and MLLM agents. Three validations are included: (1) the ratio of seed nodes correctly matching the specified seed size, (2) the ratio of seed nodes that correctly exclude non-existent nodes, and (3) the ratio of non-redundant seed nodes in each seed set.

**MLLM agents are capable of selecting seed sets that align with the specified criteria in the full-label case:** Due to the LLM hallucination (Xu et al., 2024; Duan et al., 2024) , we examine the feasibility and correctness of selected seeds by MLLM. The criteria include checking for repetitive or invalid nodes in the seed nodes and ensuring that the selected seed size meets our specifications. Table 3 shows that across three networks, the validation results are consistently high, with most metrics achieving 100% accuracy for all agents.

| Dolphins | Agent 1 | | Agent 2 | | Agent 3 | |
|---|---|---|---|---|---|---|
| | $|S| = 5$ | $|S| = 10$ | $|S| = 5$ | $|S| = 10$ | $|S| = 5$ | $|S| = 10$ |
| Validation 1 | 100.0% | 100.0% | 100.0% | 100.0% | 100.0% | 100.0% |
| Validation 2 | 100.0% | 100.0% | 100.0% | 100.0% | 100.0% | 100.0% |
| Validation 3 | 100.0% | 100.0% | 100.0% | 100.0% | 100.0% | 100.0% |
| **Lesmis** | Agent 1 | | Agent 2 | | Agent 3 | |
| | $|S| = 5$ | $|S| = 10$ | $|S| = 5$ | $|S| = 10$ | $|S| = 5$ | $|S| = 10$ |
| Validation 1 | 100.0% | 100.0% | 100.0% | 100.0% | 100.0% | 100.0% |
| Validation 2 | 100.0% | 100.0% | 100.0% | 100.0% | 100.0% | 100.0% |
| Validation 3 | 100.0% | 100.0% | 100.0% | 100.0% | 100.0% | 99.0% |
| **Polbooks** | Agent 1 | | Agent 2 | | Agent 3 | |
| | $|S| = 5$ | $|S| = 10$ | $|S| = 5$ | $|S| = 10$ | $|S| = 5$ | $|S| = 10$ |
| Validation 1 | 100.0% | 100.0% | 100.0% | 100.0% | 100.0% | 100.0% |
| Validation 2 | 100.0% | 100.0% | 100.0% | 100.0% | 100.0% | 100.0% |
| Validation 3 | 100.0% | 100.0% | 100.0% | 100.0% | 100.0% | 100.0% |

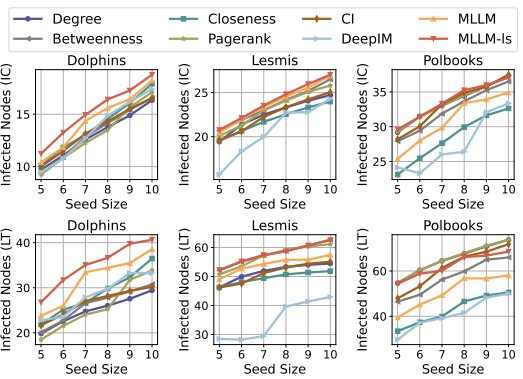

**MLLM plus local search would become a new paradigm for combinatorial optimization:** Figure 6 shows the results of IM using various strategies. In both IC and LT models, the MLLM-ls consistently outperforms other strategies, achieving a higher number of infected nodes across all seed sizes compared to traditional centrality methods such as degree, betweenness, and CI, as well as representation learning-based DeepIM, in selecting seeds for IM within networks. As shown in Figure 5, the agents' prompts are straightforward and intuitive, highlighting that MLLM is not only effective but also user-friendly, making it highly accessible for practical use.

Figure 6: The comparative IM performance on small-scale networks with the IC and LT models.

**MLLM exhibits an excellent inherent intelligence:** Figure 7 shows the distribution of infected nodes using different seed nodes suggested by different agents. The performance of the different agents across networks varies significantly due to their distinct strategies. Agent 1, which operates without specific hints, consistently performs as well as other agents with guidance across all networks. This indicates that the MLLM's capability has reached a high level of intelligence and can make optimal selections, even without explicit guidance.

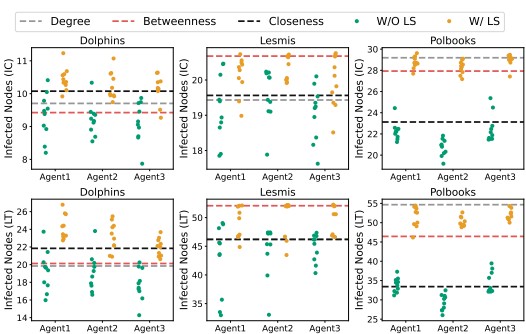

Figure 7: The IM result of MLLM agents with and without local search on small-scale networks.

**The visualization poses a challenge to MLLM for accurately recognizing the node in the dense network:** In the Polbooks network, which is both larger and denser, the visual complexity increases, making it more challenging for the agents to effectively recognize optimal seed nodes. This is where local search plays a crucial role, as demonstrated by the improvement on Polbooks as well as Dolphins and Lesmis. It helps refine the selection in a visually dense network, where visual inspection alone may not be sufficient. For the statistical results of different agents, please refer to Section A.10 in Appendix.

## 3.2 LARGE-SCALE NETWORK

The details of agents for the large-scale networks are shown in Figure 8. Due to the substantial number of nodes of large-scale networks, it is impractical to plot all the labels in a canvas of limited size. Thus, only a certain ratio of high-degree nodes of each network is displayed in the image.

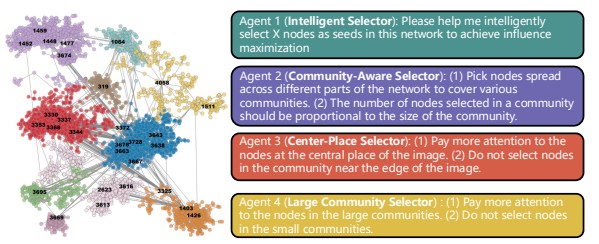

Figure 8: The illustrations of MLLM-based IM on large-scale networks. The partial-label network (left) will be inputted into MLLM along with the prompts for agents (right).

In this case, the input to MLLM becomes an image with partial labels. As seen from the prompt for agents and the input image (see Figure 19), we also include the community information compared to the full-label case. This is because (1) While MLLM demonstrates strong spatial intelligence, we still need some assistance to explicitly guide it in selecting area nodes when incorporating selection biases. (2) There is still a lack of visualization tools that effectively display the network structure globally. Thus, we utilize community detection to cluster densely connected nodes and separate loosely connected parts for better visualization. Advancements in visualization will unlock significant potential for MLLM in large-scale graph-structured problems, which will be discussed further in Section 5.

Table 4: The validations across different networks and MLLM agents. Three validations are included: (1) the ratio of seed nodes correctly matching the specified seed size, (2) the ratio of seed nodes that correctly exclude non-existent nodes, and (3) the ratio of non-redundant seed nodes in each seed set.

| Facebook | Agent 1 | | Agent 2 | | Agent 3 | | Agent 4 | |
|---|---|---|---|---|---|---|---|---|
| | $\|S\| = 10$ | $\|S\| = 20$ | $\|S\| = 10$ | $\|S\| = 20$ | $\|S\| = 10$ | $\|S\| = 20$ | $\|S\| = 10$ | $\|S\| = 20$ |
| Validation 1 | 100.0% | 100.0% | 100.0% | 100.0% | 100.0% | 100.0% | 100.0% | 100.0% |
| Validation 2 | 99.0% | 97.5% | 99.0% | 98.5% | 98.0% | 99.0% | 99.0% | 100.0% |
| Validation 3 | 100.0% | 99.5% | 100.0% | 97.5% | 100.0% | 99.5% | 100.0% | 99.0% |
| **Router** | **Agent 1** | | **Agent 2** | | **Agent 3** | | **Agent 4** | |
| | $\|S\| = 10$ | $\|S\| = 20$ | $\|S\| = 10$ | $\|S\| = 20$ | $\|S\| = 10$ | $\|S\| = 20$ | $\|S\| = 10$ | $\|S\| = 20$ |
| Validation 1 | 100.0% | 100.0% | 100.0% | 100.0% | 100.0% | 100.0% | 100.0% | 100.0% |
| Validation 2 | 98.0% | 98.5% | 99.0% | 98.5% | 98.0% | 91.5% | 98.0% | 96.0% |
| Validation 3 | 100.0% | 99.5% | 100.0% | 97.5% | 100.0% | 99.5% | 100.0% | 99.0% |
| **Sex** | **Agent 1** | | **Agent 2** | | **Agent 3** | | **Agent 4** | |
| | $\|S\| = 10$ | $\|S\| = 20$ | $\|S\| = 10$ | $\|S\| = 20$ | $\|S\| = 10$ | $\|S\| = 20$ | $\|S\| = 10$ | $\|S\| = 20$ |
| Validation 1 | 100.0% | 100.0% | 100.0% | 100.0% | 100.0% | 100.0% | 100.0% | 100.0% |
| Validation 2 | 93.0% | 85.0% | 89.0% | 88.0% | 92.0% | 91.5% | 92.0% | 80.0% |
| Validation 3 | 99.0% | 99.5% | 99.0% | 99.5% | 100.0% | 99.5% | 99.0% | 97.5% |

As seen in Table 4, the agents demonstrate strong correctness across most networks, particularly in correctly matching the specified seed size and avoiding selecting redundant nodes. As observed in Figure 19, the displayed nodes in Sex are more than the other two networks, which poses a challenge to accurately identifying the node label, reflected by the relatively low accuracy in Validation 2. A further discussion can be found in Section 5 and Figure 14(c).

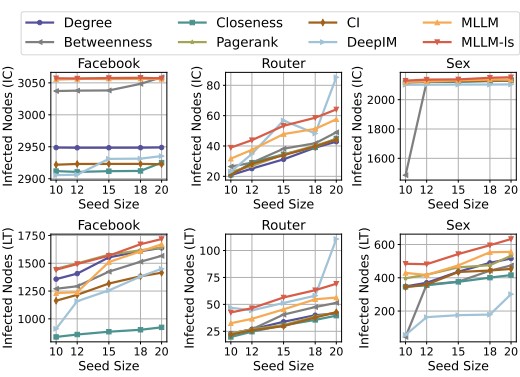

Figure 9: The comparative IM performance on large-scale networks with the IC and LT models.

**MLLM performs also well on large-scale networks**: Figure 9 presents the IM results on large-scale networks. As observed, MLMM-ls outperforms all tested methods including the state-of-the-art GNN-based DeepIM, while MLLM without local search can also surpass most centrality and hand-crafted approaches, suggesting the applicability of MLLM on real-world networks that are typically large-scale. Considering its simplicity and effectiveness, MLLM along with basic optimization techniques will be a promising candidate for large-scale graph problems.

Figure 10 shows the IM results of different agents on large-scale networks. In several cases, the MLLM agents, particularly Agent 1, outperform traditional centrality-based methods such as degree or betweenness centrality, even when local search is not applied. This suggests that MLLMs have an inherent capability to select influential nodes even without being explicitly directed, rivaling or exceeding conventional metrics that rely on predefined structural properties.

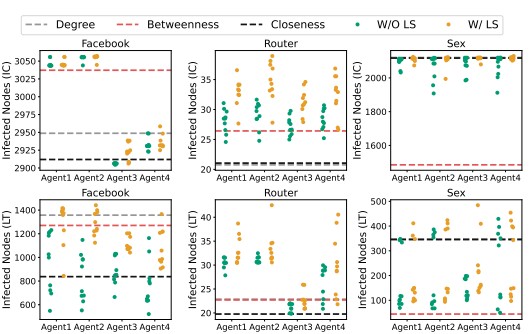

Figure 10: The IM result of MLLM agents with and without local search on large-scale networks.

**Mixed agents with different strategies can be easily adapted to various scenarios:** The variation in performance across different networks, as seen with Agent 3 being the worst performer in the Router network but the best in the Sex network, suggests that different agents are better suited for specific types of network topologies. This observation implies that no single strategy is universally optimal across all scenarios. A combination of agents with different selection biases could provide a more robust and adaptable approach, leveraging the strengths of each agent based on the network's unique structure. It is to be expected that more sophisticated agents will achieve better performance in the future.

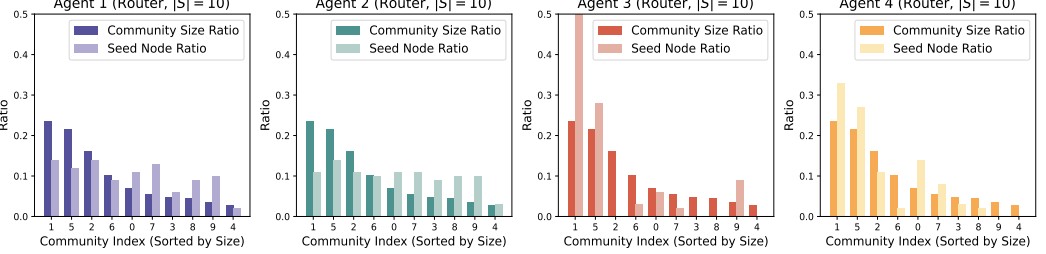

Figure 11: The distribution of ten selected seed nodes by MLLM agents on the Router network. The community size ratio (darker bars) refers to the proportion of a given community's size relative to the total size of the entire network. The seed node ratio (lighter bars) refers to the proportion of seed nodes selected from a specific community relative to the total number of seed nodes.

**MLLM exhibits an excellent spatial awareness:** Figure 11 presents the distribution of sampled nodes by four different MLLM agents in the Router network, each with a seed size of 10. MLLM exhibits spatial intelligence, as seen in Agent 1, which operates without specific guidance yet still

distributes seed nodes in a balanced manner across communities. Furthermore, the results show that the MLLM agents can accurately follow the specific guidance provided to them. For example, Agent 2, tasked with distributing nodes proportionally across communities, adheres closely to the community size ratio. The results also reveal that certain agents, such as Agents 3 and 4, rarely select any seed nodes from certain communities. This is particularly evident in smaller communities where these agents' biases led them to focus primarily on larger or more central communities. Agent 3, with its emphasis on central nodes in the image, and Agent 4, which prioritizes large communities, both completely overlooked some of the smaller communities in the network.

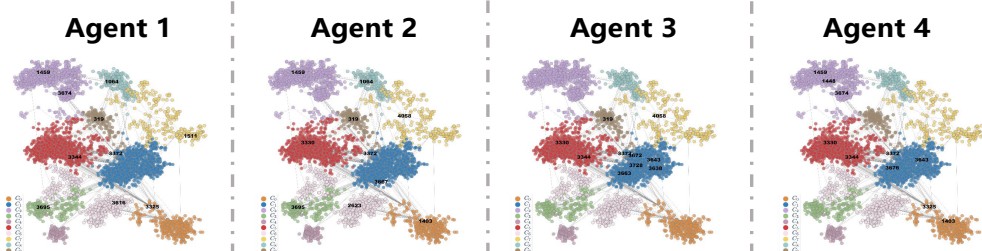

Figure 12: The seeds selected by different agents on Router. Agent 1: Intelligent Selector; Agent 2: Community-Aware Selector; Agent 3: Center-Place Selector; Agent 4: Large Community Selector.

**MLLM possesses the deep understanding of graph problems without any fine-tuning:** Some selection seeds of the different agents are shown in Figure 12 (see Section A.11 for full selection results). Agent 1 takes into account both the diversity of the selection area and the avoidance of selecting nodes from small and peripheral communities (as can be seen from the low seed node ratios in communities 4 in Figure 11). These aspects are exactly the core idea of Agents 2, 3 and 4, which are guided by humans.

# 4  MLLM ON BASIC GRAPH-RELATED TASKS

In this section, we will investigate the MLLM on some basic graph-structured tasks and identify factors affecting the performance of MLLM.

Table 5: Structural Metrics of Synthetic Networks: Including Average Node and Edge Counts, Degree, Shortest Distance, Connected Components, and Cycle Presence Proportion.

Three types of random networks are utilized: Barabási-Albert (BA) network, Erdős-Rényi (ER) network and Watts-Strogatz (WS) network. Table 5 lists the structural information of these networks where BA is viewed as dense network and WS and ER are relatively sparse sometimes with containing multiple connected components.

| Metrics | WS | | BA | | ER | |
|---|---|---|---|---|---|---|
| | Easy | Hard | Easy | Hard | Easy | Hard |
| Avg. Nodes | 7.58 | 17.58 | 7.69 | 17.51 | 12.63 | 17.39 |
| Avg. Edges | 7.58 | 17.58 | 12.38 | 32.01 | 15.01 | 14.10 |
| Avg. Degree | 2.00 | 2.00 | 3.18 | 3.65 | 2.33 | 1.61 |
| Avg. Shortest Dist. | 1.61 | 1.57 | 1.547 | 2.08 | 0.82 | 0.11 |
| Avg. Component | 1.27 | 1.83 | 1.00 | 1.00 | 2.20 | 5.15 |
| Cycle Existence | 100.0% | 100.0% | 100.0% | 100.0% | 100.0% | 100.0% |

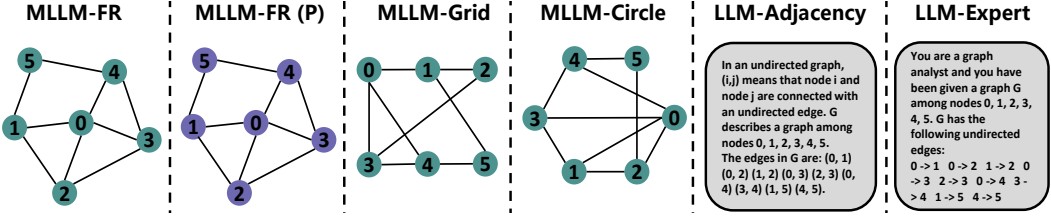

Figure 13: Illustrations of various representations of the same graph with six nodes, including input images with different layouts and colors provided to the MLLM, accompanied by the sentence, "You are an expert in network science and will be provided with a network G in the form of an image," along with two types of textual descriptions.

**MLLM excels in tasks requiring global awareness:** The performance of MLLM-FR and MLLM-FR (P) in tasks 3 (Highest Betweenness Node), 4 (Shortest Distance), and 6 (Connected Components)

Table 6: The capability of different models on the basic graph-structured task. Task 1 (Node Degree): Calculate the degree of a specific node; with the highest betweenness centrality; Task 2 (Highest Degree Node): Identify the node with the highest number of connections; Task 3 (Highest Betweenness Node): Identify the node Task 4 (Shortest Distance): Determine the shortest path between two specified nodes; Task 5 (Cycle Detection): Identify whether the network contains a cycle. Task 6 (Connected Components): Identify the number of distinct connected components.

| Model | Task 1 | | Task 2 | | Task 3 | | Task 4 | | Task 5 | | Task 6 | |
|---|---|---|---|---|---|---|---|---|---|---|---|---|
| | Easy | Hard | Easy | Hard | Easy | Hard | Easy | Hard | Easy | Hard | Easy | Hard |
| LLM-Expert | 89.5% | 74.0% | 98.5% | **92.5%** | 72.0% | 72.5% | **89.5%** | **58.0%** | 100.0% | 100.0% | 100.0% | 100.0% |
| LLM-Adjacency | **96.0%** | **76.5%** | **99.5%** | 91.0% | 71.0% | **75.5%** | 86.0% | 51.0% | 100.0% | 100.0% | 100.0% | 99.5% |
| MLLM-FR | 54.5% | 36.0% | 88.5% | 77.5% | 77.5% | 69.0% | 62.5% | 39.0% | 100.0% | 100.0% | 100.0% | 100.0% |
| MLLM-FR(P) | 63.0% | 42.5% | 88.5% | 79.5% | **80.0%** | 66.5% | 60.5% | 40.5% | 100.0% | 100.0% | 100.0% | 100.0% |
| MLLM-Circle | 19.0% | 11.5% | 91.5% | 59.0% | 75.5% | 53.5% | 63.0% | 45.0% | 99.5% | 100.0% | 100.0% | 100.0% |
| MLLM-Grid | 26.0% | 10.0% | 64.0% | 25.5% | 43.0% | 16.5% | 53.0% | 48.5% | 100.0% | 100.0% | 98.5% | 99.5% |

(a) Barabási-Albert (BA) network. The number of edges each new node connects to when it is added to the network is set to 2. #Easy: $n \in [5, 10]$; #Hard: $n \in [15, 20]$.

| Model | Task 1 | | Task 2 | | Task 3 | | Task 4 | | Task 5 | | Task 6 | |
|---|---|---|---|---|---|---|---|---|---|---|---|---|
| | Easy | Hard | Easy | Hard | Easy | Hard | Easy | Hard | Easy | Hard | Easy | Hard |
| LLM-Expert | 88.0% | 94.0% | **91.5%** | 90.0% | 59.5% | 64.0% | 64.0% | 66.5% | 74.0% | 49.0% | 23.5% | 26.0% |
| LLM-Adjacency | 95.5% | **94.5%** | 91.0% | **94.5%** | 62.5% | 70.0% | 60.5% | 60.5% | 93.0% | 82.5% | 32.0% | 26.0% |
| MLLM-FR | 76.0% | 81.5% | 81.5% | 84.0% | **68.0%** | 67.0% | **65.0%** | **67.0%** | 86.0% | 75.5% | **93.0%** | **54.5%** |
| MLLM-FR(P) | 73.0% | 82.0% | 80.0% | 91.0% | 65.0% | **77.0%** | 52.0% | 61.5% | 89.0% | 72.0% | 87.0% | **54.5%** |
| MLLM-Circle | 21.5% | 12.5% | 73.0% | 72.0% | 44.0% | 45.5% | 32.0% | 15.5% | **97.0%** | **97.0%** | 43.5% | 5.0% |
| MLLM-Grid | 19.5% | 17.5% | 49.0% | 47.0% | 20.0% | 24.5% | 34.5% | 22.5% | **99.0%** | 97.0% | 45.5% | 4.5% |

(b) Erdős-Rényi (ER) network. The probability that any pair of nodes will have an edge connecting them is set to 0.2 for the easy case and 0.1 for the hard case. #Easy: $n \in [10, 15]$; #Hard: $n \in [15, 20]$.

| Model | Task 1 | | Task 2 | | Task 3 | | Task 4 | | Task 5 | | Task 6 | |
|---|---|---|---|---|---|---|---|---|---|---|---|---|
| | Easy | Hard | Easy | Hard | Easy | Hard | Easy | Hard | Easy | Hard | Easy | Hard |
| LLM-Expert | **98.5%** | 94.5% | **99.0%** | 92.5% | 78.0% | 43.0% | 76.5% | 47.0% | 84.0% | 92.5% | 57.5% | 29.5% |
| LLM-Adjacency | 95.5% | **95.5%** | **99.0%** | **98.5%** | 73.0% | 53.5% | **80.0%** | 36.0% | 100.0% | 100.0% | 70.5% | 33.0% |
| MLLM-FR | 81.5% | 66.5% | 96.5% | 82.0% | **90.0%** | 57.0% | 69.0% | 47.0% | 91.0% | 90.0% | 99.0% | 89.5% |
| MLLM-FR(P) | 77.5% | 74.0% | 97.5% | 88.5% | 88.5% | **68.0%** | 58.5% | **50.0%** | 89.5% | 85.0% | **100.0%** | **93.5%** |
| MLLM-Circle | 64.5% | 47.5% | 90.5% | 70.5% | 70.5% | 32.0% | 52.5% | 26.0% | 98.0% | **100.0%** | 93.0% | 43.0% |
| MLLM-Grid | 27.5% | 21.0% | 63.5% | 50.0% | 43.5% | 20.0% | 49.5% | 23.0% | 97.0% | 98.5% | 85.5% | 50.5% |

(c) Watts-Strogatz (WS) network. The number of nearest neighbors each node is connected to in the initial ring lattice is set to 1 and the probability of rewiring each edge is set to 0.2. #Easy: $n \in [5, 10]$; #Hard: $n \in [15, 20]$.

showcases their ability to handle problems that require a comprehensive understanding of the entire network structure. MLLM's ability to process these global relationships efficiently leads to its dominance over other methods in such tasks.

**The color has minimal impact on MLLM's performance:** The close similarity in results between MLLM-FR and MLLM-FR (P) demonstrates that the color visual representations has little influence on the model's effectiveness since both layouts provide nearly identical performance across the tasks.

**Layout significantly affects performance:** The difference between the results of MLLM-FR and models using MLLM-Circle or MLLM-Grid layouts highlights the importance of the layout. MLLM-FR, which uses a force-directed layout, provides clearer visual cues of the network's structure, leading to superior performance. In contrast, MLLM-Circle and MLLM-Grid offer less intuitive spatial arrangements, making it harder for the model to recognize global features, which leads to poorer results across tasks. Moreover, some layouts will even lost some basic structural information, for example, the connection of node 0 and node 2 cannot reflected in the grid case of Figure 13.

**MLLM's adaptability across different network structures:** MLLM maintains performance in global tasks (3 and 6) regardless of network density, as evidenced by its comparable results in both sparse networks like ER and WS and denser networks like BA. In contrast, LLM shows a marked drop in performance, particularly in sparser networks, where spatial awareness is crucial for success. MLLM's ability to retain its effectiveness across these varying structures highlights its suitability for tasks that require a broader perspective, where LLM struggles due to its localized understanding.

**MLLM's strength over LLM in large-scale problems:** The superior performance of MLLM in tasks requiring global awareness suggests that it is better equipped to handle large-scale problems

where a comprehensive understanding of the entire network is essential. Furthermore, LLM's reliance on extensive natural language prompts when encoding large-scale graphs further limits its capability, making MLLM a more suitable choice for tasks that involve larger, more complex network structures.

## 5 DISCUSSION AND PROSPECT

In addition to the aforementioned spatial intelligence of MLLMs on graph-structured problems, another key strength of MLLMs lies in their remarkable scalability, which is particularly advantageous when dealing with large-scale networks. Real-world networks are typically massive (Leskovec & Sosič, 2016), making it impractical to encode the entire network into a text-based prompt. In contrast, by leveraging visual inputs in the form of network images, MLLMs bypass this limitation. Regardless of how large or complex the network is, the input remains a fixed-size image, allowing the MLLM to interpret and process it efficiently. Unlike adjacency matrices and learned embeddings, which trade off structural information for computation, images serve as the most intuitive representation of graph structures, effectively preserving valuable high-order information such as community structures, paths, and motifs, and so on.

The current MLLMs may sometimes return undesirable outcomes. Figure 14 shows several possible recognition results of MLLMs on one graph. The original graph consists of three nodes (1, 2, and 3) where node 1 is connected to node 2, and node 2 is connected to node 3. Case (a): This is the correct recognition of the graph by the MLLMs. Case (b): The MLLMs incorrectly recognize the structure by displaying node 2 between node 1 and node 3 but fail to recognize the edge between nodes 1 and 2. Case (c): In this scenario, nodes 1 and 2 are so close to each other that the MLLMs misrecognize them as a single node labeled '12'.

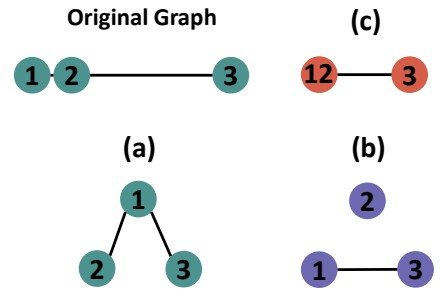

Figure 14: An example of possible outcomes from MLLM recognition on the same graph.

As observed, MLLMs' full potential is still constrained by the lack of effective visualization tools. This is the reason why we call this representation of graph as low-loss. Even humans face difficulties in recognizing and interpreting individual nodes when a large number of them are plotted on a fixed-size canvas. In such cases, a tool analogous to a magnifying glass would allow for a more detailed, micro-level examination of specific areas of the network. This limitation in visualization should not be considered a flaw in MLLMs itself, as it reflects a broader challenge in rendering and interpreting complex, dense networks visually.

If visualization software can be seamlessly integrated with MLLMs to support interactive exploration—enabling zooming and detailed node examination in real-time, the performance and applicability of MLLMs would be greatly enhanced. This would not only improve MLLMs' reasoning capabilities on large-scale networks but also enable full-scale labeling and analysis, similar to what is currently achievable with small-scale networks. Achieving this would allow for loss-free representation of graph-structured data through images, opening a new paradigm for graph-related computations. Note that the proposed MLLM-based method are generalizable and could extend beyond the problems studied here to other challenges, such as graph coloring, vertex cover, and graph partitioning, with our present work providing a strong foundation for these future developments.

## 6 CONCLUSION

In this work, we have demonstrated the effectiveness of MLLMs in addressing complex graph-structured problems, such as network dismantling and influence maximization. By utilizing simple prompts combined with local search strategies, our approach achieves superior performance over traditional methods and GNN-based approaches. We provided a comprehensive analysis of MLLMs' capabilities on fundamental graph tasks and identified key factors that enhance their effectiveness. Our findings reveal the potential of MLLMs to revolutionize large-scale graph problem-solving, marking a significant step toward harnessing their full capacity in practical, real-world applications.

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

# A   APPENDIX

In this section, we will provide additional information regarding our work as follows:

- Section A.1 reviews related work on influence maximization, network dismantling, and the application of LLMs and MLLMs on graph-structured tasks.

- Section A.2 introduces three spreading models used in our work.

- Section A.3 introduces the experimental setting of our work.

- Section A.4 is our proposed visualization method including

- Section A.5 presents our proposal local search method.

- Section A.6 lists the benchmarks regarding influence maximization and network dismantling.

- Section A.7 gives the prompts of different agents adapted to different tasks.

- Section A.8 introduces three random works tested in Section 4, including Barabási-Albert (BA) network, Erdős-Rényi (ER) network and Watts-Strogatz (WS) network.

- Section A.9 shows the results of different methods on the SI spreading model.

- Section A.10 shows the statistical results of different agents regarding IC and LT models.

- Section A.11 presents the result of the distribution of selected seeds.

## A.1   RELATED WORK

### A.1.1   INFLUENCE MAXIMIZATION

Influence maximization is a computational problem in network science where the goal is to identify a set of key nodes in a network that maximizes the spread of information, behavior, or influence through the network. By setting a predefined diffusion model, the greedy algorithm (Kempe et al., 2003) was employed to iteratively identify the node with the largest influence spread until the desired network size was achieved. Although greedy algorithms can achieve excellent performance but it involves a huge number of simulatation of spreding process, limiting its applicability on large-scale data. The Cost-Effective Lazy Forward (CELF) algorithm (Leskovec et al., 2007) was then proposed significantly reduces computational complexity by leverage the submodularity of the influence function to avoid unnecessary recalculations.

Heuristic methods in influence maximization offer a balance between simplicity and effectiveness, making them attractive for large-scale network applications where computational resources are limited. The representative heuristic methods includes degree centrality (Freeman et al., 2002), be-tweenness centrality (Freeman, 1977), closeness centrality (Wasserman, 1994), eigenvalues (Berman & Plemmons, 1994), pagerank (Page, 1999), k-core (Batagelj & Zaversnik, 2003) and so on.

On the other hand, several meta-heuristics have been proposed based on different bio-inspired evolu-tionary techniques to solve this complex combinatorial problem due to their flexible representation of solutions and effectiveness. Gong *et al.* proposed a particle swarm optimization (Gong et al., 2016b) to search for the optimal seed. Other techniques are also explored in this task, such as ant colony (Salavati & Abdollahpouri, 2019), memetic algorithm (Gong et al., 2016a) and differential evolution (Li et al., 2022). As evolutionary optimization usually relies the population, there are also several work focusing on improving the quality of initialized population, such as (Zhang et al., 2022; Qiu et al., 2021).

Due to the representation capabilities of graph neural networks on graph structures, researches has shifted from classical tasks like node classification to combinatorial optimization. Yu *et al.* transformed the influence maximization problem into a regression problem by representing adjacency matrix into embeddings using GNNs (Yu et al., 2020). From the perspective of critical nodes identification, Ma *et al.* studied the adversarial attack on GNNs based on linear threshold model. Recently, Ling *et al.* proposed DeepIM (Ling et al., 2023), aiming to learn the latent representation of the seed nodes in an end-to-end training manner.

### A.1.2 Network Dismantling

Network dismantling involves identifying the minimal group of nodes whose removal most rapidly leads to the network's fragmentation, as outlined in the optimal percolation problem (Artime et al., 2024). This task focuses on isolating nodes that, when eliminated, quickly reduce the largest connected component (LCC) to disjoint sub-groups, thereby impairing the network's operational capacity. A straightforward approach involves targeting nodes based on their centrality measures, with the node degree being a primary metric. This method targets highly connected nodes or hubs under the premise that nodes with more connections are more influential (Albert et al., 2000; Cohen et al., 2001). Various other heuristic measures of centrality are also applicable for pinpointing these critical nodes. Moreover, these strategies can be categorized into static and dynamic types. Static strategies determine the sequence of node (or edge) removal at the start of the dismantling process, while dynamic strategies adjust this sequence as the network's structure evolves during the process.

Drawing inspiration from decycling-based techniques, CoreHD focuses on decycling a network by sequentially removing the highest-degree nodes within the 2-core (Zdeborová et al., 2016). Another approach, known as explosive immunization has been introduced by considering explosive percolation (EP) with strategies to keep network clusters highly fragmented. Generalized network dismantling, meanwhile, tackles the variable costs associated with node removals by iteratively eliminating nodes that optimally impact an approximate spectral partitioning. Additionally, there have been advancements in applying machine learning to network attacks, such as graph dismantling with machine learning (GDM) (Grassia et al., 2021) and FINDER (Fan et al., 2020). GDM casts this network dismantling problem into a regression problem and employs a supervised learning framework training on networks pre-dismantled. FINDER is a reinforcement larning-based framework training on a large number of small-scale networks and then generalized to the real-world networks.

### A.1.3 LLMs and MLLMs in graph-structured problems

LLMs have proven effective in many areas, leading to the question of their applicability to graph-structured data. Chen *et al.* employed LLMs as an enhancer and a predictor, respectively (Chen et al., 2024). The LLM-based enhancer augments node features, while the LLM-based predictor directly outputs the classification. A model combining LLMs and graph learning method named GraphLLM was proposed (Chai et al., 2023) to enhance the accuracy reasoning tasks on the text-attributed graphs (TAGs).

However, TAGs are not prevelent as it is challenging to build the label and textual feature for a huge number of nodes. Thus, these LLM-based work is still not enough tackle the real-world problems where there is only structural information available. Thus, some studies sought to directly encode graph structures into text through different prompt engineering techniques (Fatemi et al., 2023; Wang et al., 2024), enabling LLMs to comprehend and analyze these structures. However, experimental results show that LLMs have significantly limited reasoning capabilities, even with small-scale networks, let alone large-scale real-world networks.

There are a few work using MLLMs on combinatorial problems. Huang *et al.* used visual and text information to solve the traveling salesman problem (TSP) (Huang et al., 2024). In the following, Elhenawy *et al.* proposed finding the optimal route with graphical data solely and tested the effectiveness of different MLLMs (Elhenawy et al., 2024). However, the current work has only verified the limited feasibility of MLLMs in combinatorial optimization, and there remains a significant gap before practical application can be achieved. Firstly, the datasets employed in these studies are relatively small, containing at most fewer than 200 nodes. Secondly, the optimization outcomes do not compare favorably with commonly used benchmarks, indicating that the potential of MLLMs has not been fully realized.

### A.2 Spreading models

The effectiveness of influence maximization of different methods is examined with three spreading models Independent Cascade model (Robson et al., 2024), Linear Threshold model (Riquelme et al., 2018) and Susceptible-Infected model (Zhao & Cheong, 2024).

**Independent Cascade (IC) model**: It is a diffusion model used to simulate the spread of influence in a network. In this model, each activated node has a single chance to activate each of its inactive

neighbors with a given probability $p$. If node $u$ becomes active at time $t$, it will attempt to activate each inactive neighbor $v$ at time $t + 1$. The process continues until no more activations are possible.

**Linear Threshold (LT) model**: It is another diffusion model that assumes each node in the network has a threshold $\theta_v \in [0, 1]$. A node $v$ becomes active if the sum of the influences from its active neighbors exceeds its threshold. Each edge $(u, v)$ has an associated weight $w_{uv}$ such that $\sum_{u \in N(v)} w_{uv} \leq 1$, where $N(v)$ is the set of neighbors of $v$. The activation condition for node $v$ is:

$$\sum_{u \in N(v), \text{active}} w_{uv} \geq \theta_v$$

.

**Susceptible-Infected (SI) model**: It is a simple epidemiological model where nodes can be in one of two states: susceptible (S) or infected (I). Assume $v$ is a susceptible node at time $t \in \mathbb{N}^+$. If infected nodes surround node $v$, the probability that node $v$ will become infected at time $t + 1$ can be determined as:

$$\mathbf{P}(v, t + 1) = 1 - \prod_{v \in \mathcal{N}_I(v)} (1 - \mathbf{P}_{uv}), \tag{1}$$

where $\mathcal{N}_I(v)$ denotes the node $v$ neighbors that have been infected and $\mathbf{P}_{uv}$ denotes the likelihood of $u$ infecting $v$.

## A.3 EXPERIMENTAL SETTING

As our work is not aiming to compare the performance of MLLMs but to explore an novel solution to graph tasks, we directly select the state-of-the-art model *gpt-4o-2024-08-06* as our backbone. In network dismantling, the agent makes 20 attempts on each network. For influence maximization, we design 4 agents for the partial-label case and 3 agents for the full-label case, with each agent sampling nodes 10 times. In the validation, we use Monte Carlo methods to simulate 100,000 spreading processes for the IC and LT models, and 100 times for the SI model. The infection probability of SI model and IC model is set to 0.1. For better visualization, we merge some of communities in large-scale network and the related information can be found in Table 7 where the number of merged communities is experimentally obtained.

Table 7: The number of original communities and the merged communities.

| Network | Original | Merged |
|---|---|---|
| Facebook | 13 | 9 |
| Router | 63 | 9 |
| Sex | 170 | 10 |

## A.4 VISUALIZATION METHOD

In the analysis of large networks, the detection and analysis of communities is crucial. However, when applying standard community detection algorithms like Fastgreedy (Clauset et al., 2004), the number of communities detected can often exceed practical utility, especially in large networks. These algorithms tend to identify many small communities that may be of less relevance or too granular for specific applications. Consequently, there is a need for a method to merge these smaller communities into larger, more meaningful groups.

This paper presents an algorithm designed to merge small communities into fewer, larger communities while maintaining the integrity and connectivity of the original network structure. The goal is to reduce the number of communities to a more manageable size, aligning with user-defined requirements or specific analytical needs.

The algorithm receives a graph $G$, an initial set of communities $C$, and a target number of communities $T$. The goal is to merge smaller communities into their nearest neighbors until the number of communities is reduced to $T$.

1. **Identify the Smallest Community**: In each iteration, the algorithm identifies the smallest community by comparing the sizes of all communities.

2. **Count Edges to Other Communities**: For each edge in the graph, the algorithm checks if the edge connects the smallest community to any other community. It keeps track of how many edges each neighboring community has connected to the smallest community.

3. **Find the Closest Community**: The community with the highest number of edges connected to the smallest community is chosen as the "closest" community.

4. **Merge Communities**: All nodes in the smallest community are reassigned to the closest community. The indices of the other communities are adjusted accordingly to reflect the reduction in the number of communities.

5. **Repeat**: This process continues until the number of communities equals the target number.

The visualization of networks has different layouts. In this work, we have tested three types to investigate the influence of layouts on the effectiveness of MLLMs.

- **Fruchterman-Reingold Layout**: The Fruchterman-Reingold (FR) layout is a force-directed algorithm that simulates physical forces between the nodes and edges of a graph. Nodes repel each other like charged particles, while edges act like springs that pull connected nodes together. The goal is to position the nodes in a way that minimizes edge crossings and evenly distributes them, creating an aesthetically pleasing and clear representation of the network.

- **Circle Layout**: In the Circle layout, all nodes are placed at equal distances from each other along the circumference of a circle. This layout is useful when the relationships between nodes are not hierarchical or when you want to emphasize the circular arrangement. It is a simple and symmetric way to visualize a network, making it easy to see the overall structure.

- **Grid Layout**: The Grid layout arranges nodes in a regular grid pattern, with each node occupying a unique position. This layout is effective for displaying nodes in a structured, non-overlapping manner, making it easier to compare their positions and relationships. It's often used when clarity and simplicity are priorities, especially in networks where the spatial arrangement of nodes is important.

### A.5 LOCAL SEARCH

Given a graph $\mathcal{G} = (\mathcal{V}, \mathcal{E})$, where $\mathcal{V}$ represents the set of nodes and $\mathcal{E}$ denotes the set of edges, the objective is to identify a set of seed nodes $S \subset \mathcal{V}$ that maximizes the influence spread throughout the network. The influence spread is evaluated using a predefined influence diffusion model, such as the Independent Cascade (IC), Linear Threshold (LT) and Susceptible-Infected (SI) models. For the sake of efficiency, the iteration number is set to 5 and the simulation number of spreading process is 5,000.

The algorithm terminates after a predefined number of iterations, ensuring that the search process is controlled and does not continue indefinitely.

### A.6 BENCHMARKS

#### A.6.1 INFLUENCE MAXIMIZATION

**Degree** measures the number of direct connections a node has. For a node $v$, degree centrality $\deg(v)$ is given by:

$$\deg(v) = \sum_{u \in V} a_{vu},$$

where $a_{vu}$ is the element of the adjacency matrix indicating the presence of an edge between nodes $v$ and $u$.

**Betweenness** measures the extent to which a node lies on the shortest paths between other nodes. For a node $v$, betweenness centrality $\text{BC}(v)$ is given by:

$$\text{BC}(v) = \sum_{s \neq v \neq t} \frac{\sigma_{st}(v)}{\sigma_{st}},$$

where $\sigma_{st}$ is the total number of shortest paths from node $s$ to node $t$, and $\sigma_{st}(v)$ is the number of those paths that pass through $v$.

---

**Algorithm 1** Local Search Algorithm for Influence Maximization

---

1: **Precompute** degrees and betweenness for all nodes in the graph.
2: **Initialize** seed set $S$ with MLLM to initial_seeds.
3: **Evaluate** the initial influence spread of $S$ based on the model, stored in best_spread.
4: **for** max_iter iterations **do**
5:     improved ← False
6:     **for** each node $v$ in $S$ **do**
7:         $N(v)$ ← list of neighbors of $v$.
8:         Sort $N(v)$ based on degree or betweenness, randomly chosen.
9:         $u$ ← top-ranked neighbor not in $S$.
10:        $S' \leftarrow (S \setminus \{v\}) \cup \{u\}$.
11:        Calculate new_spread for $S'$ using the selected model.
12:        **if** new_spread > best_spread **then**
13:            $S \leftarrow S'$.
14:            best_spread ← new_spread.
15:            improved ← True.
16:            **break**
17:        **end if**
18:     **end for**
19:     **if** not improved **then**
20:        **break**
21:     **end if**
22: **end for**

---

**Closeness** measures how close a node is to all other nodes in the network. For a node $v$, closeness centrality $CC(v)$ is given by:

$$CC(v) = \frac{1}{\sum_{u \in V} d(v, u)},$$

where $d(v, u)$ is the shortest path distance between nodes $v$ and $u$.

**PageRank** measures the influence of a node based on the idea that connections to high-scoring nodes contribute more to the score of the node. For a node $v$, PageRank $PR(v)$ is given by:

$$PR(v) = \frac{1 - \alpha}{|V|} + \alpha \sum_{u \in N_{(v)}} \frac{PR(u)}{Out(u)},$$

where $\alpha$ is a damping factor and $N_{(v)}$ is the neighbors of node $v$.

**Collective influence (CI)** is based on the idea that the influence of a node within a network is not only determined by its local properties, such as its degree, but also by its position within the larger network structure (Morone & Makse, 2015). The CI of a node at a distance $l$ is calculated by considering the node's degree and the degrees of nodes that are $l$ steps away from it. Specifically, the CI of a node $v$ in a network is defined as:

$$CI_l(v) = (k_v - 1) \sum_{u \in \partial B_l(v)} (k_u - 1)$$

where $k_v$ is the degree of the node $v$ and $\partial B_l(v)$ represents the set of nodes that are exactly $l$ steps away from $v$ (the boundary of the ball of radius $l$ around $v$). $k_u$ is the degree of a node $u$ in the boundary set.

**DeepIM** is a state-of-the-art framework based on graph neural networks (GNNs) that models the seed set's representation within a latent space. This representation is concurrently trained with a model that comprehends the fundamental network diffusion mechanism with end-to-end training approach (Ling et al., 2023).

**MLLM** refers to the best seed nodes among all the attempts by agents.

**MLLM-ls** refers to the best seed nodes among all the attempts after local search by agents.

### A.6.2 Graph dismantling

**High-degree (HD):** this method involves repeatedly identifying and removing the node with the highest degree in the remaining network. This process is dynamic, as the degree of nodes changes after each removal, ensuring that the most connected node at each step is eliminated.

**High-collective influence (HCI):** Similar to HD, where at each step, the node with the highest collective influence in the remaining network is removed.

**MLLM** refers to the average result over multiple attempts of agent.

**MLLM-best** refers to the best result among multiple attempts of agent.

### A.7 Prompt engineering

In this work, our prompt to MLLM along with the image following the same format, consisting of context-setting prompt, task description and the output directive prompt. The task description of dismantling can be found in Figure 2 and the task description of influence maximization can be found in Figures 5 and 8 for partial-label case and full-label case, respectively. For the context-setting prompt and the output directive prompt, please refer to Table 8.

Table 8: Context-setting and output directive prompts for different network tasks. Context-setting is placed at the beginning of the prompt to explain the input information and the played role to agents and the output is placed at the end of the prompt to restrict the output format.

| Task | Context-setting prompt | Output directive prompt |
|------|------------------------|-------------------------|
| **Graph dismantling** | You are an expert in network science and you will be provided with a network in the form of an image. Each node is labeled with its node id in black text. | Do NOT output any other text or explanation. Just tell me the node id only. Your answer should be: node id. |
| **Influence maximization (full label)** | You are an expert in network science and will be provided with one network in the form of an image. | Do NOT output any other text or explanation. Just tell me the node IDs only. Your answer should be only a list as [node_id, ..., node_id] |
| **Influence maximization (partial label)** | You are an expert in network science and will be provided with one network in the form of an image. The network is divided into different communities and the nodes in the same community are of the same color. | Do NOT output any other text or explanation. Just tell me the node IDs only. Your answer should be only a list as [node_id, ..., node_id] |

To further assess the understanding capabilities of MLLMs on graph structures, we evaluate them on six fundamental graph problems. In addition to the image and leading sentence (see Figure 13), the question itself is also included as part of the prompt, as detailed in Table 9.

### A.8 Synthetic network

**Erdős-Rényi (ER)** network model, introduced by Paul Erdős and Alfréd Rényi, is a foundational concept in random graph theory (Erdos et al., 1960). In an ER network, a graph is constructed by connecting nodes randomly with a given probability $p$. This means each pair of nodes has an equal and independent chance of being connected by an edge. The simplicity of this model allows for easy analysis and provides insights into the properties of random graphs, such as the emergence of a giant component and phase transitions.

**Barabási-Albert (BA)** network model, proposed by Albert-László Barabási and Réka Albert, generates scale-free networks characterized by a power-law degree distribution (Barabási & Albert, 1999). This model captures the preferential attachment mechanism, where new nodes are more likely to connect to existing nodes with higher degrees. The BA model explains the emergence of hubs, or

Table 9: The prompt used to evaluate MLLMs' capabilities on six fundamental graph-structured problems.

| Problem | Question |
|---|---|
| **Node Degree** | Given the network G provided, please answer the following question: How many connections does node 1 have? The answer is a number, denoted as A1. Your output should be a list as [A1] without any text and explanation. |
| **Highest Degree Node** | Given the network G provided, please answer the following question: Which node has the highest degree value? The answer is a number, denoted as A1. Your output should be a list as [A1] without any text and explanation. |
| **Highest Betweenness Node** | Given the network G provided, please answer the following question: Which node has the highest betweenness value? The answer is a number, denoted as A1. Your output should be a list as [A1] without any text and explanation. |
| **Shortest Distance** | Given the network G provided, please answer the following question: What is the shortest distance between node 1 and node 2? The answer is a number or False if they cannot reach each other, denoted as A1. Your output should be a list as [A1] without any text and explanation. |
| **Cycle Detection** | Given the network G provided, please answer the following question: Does the network contain a cycle? The answer is either True or False, denoted as A1. Your output should be a list as [A1] without any text and explanation. |
| **Connected Components** | Given the network G provided, please answer the following question: How many connected components does the network have? The answer is a number, denoted as A1. Your output should be a list as [A1] without any text and explanation. |

highly connected nodes, which are a hallmark of many complex networks, such as the internet, social networks, and biological systems. This model provides a more realistic representation of network growth and connectivity compared to random graphs.

**Watts-Strogatz (WS)** network model, developed by Duncan J. Watts and Steven Strogatz, is designed to capture the small-world properties of real-world networks, which exhibit high clustering and short average path lengths (Watts & Strogatz, 1998). The WS model starts with a regular ring lattice where each node is connected to $k$ nearest neighbors. Then, with a probability $p$, each edge is randomly rewired, introducing shortcuts that reduce the path lengths between nodes. This model successfully balances the regularity of lattices with the randomness of completely random graphs, making it suitable for studying phenomena such as the spread of diseases and information in social networks.

A.9    RESULTS OF INFLUENCE MAXIMIZATION IN SI MODEL

Figures 15 and 16 shows the spreading speed of different methods based on the SI model, from which we can see that MLLM can achieve the faster spreading than other centralities regardless of small-scale or large-scale networks. This result indicates that MLLM is robust to the spreading model with the aid of local search. To have a more intuitive comparison, the AUC regarding to these two figures are listed in Table 10 where the larger AUC means the better performance. As seen, the size of the network poses a challenge to the MLLMs in the full-label case as MLLMs without local search is not comparable to other methods, suggesting the significance of visualization tools.

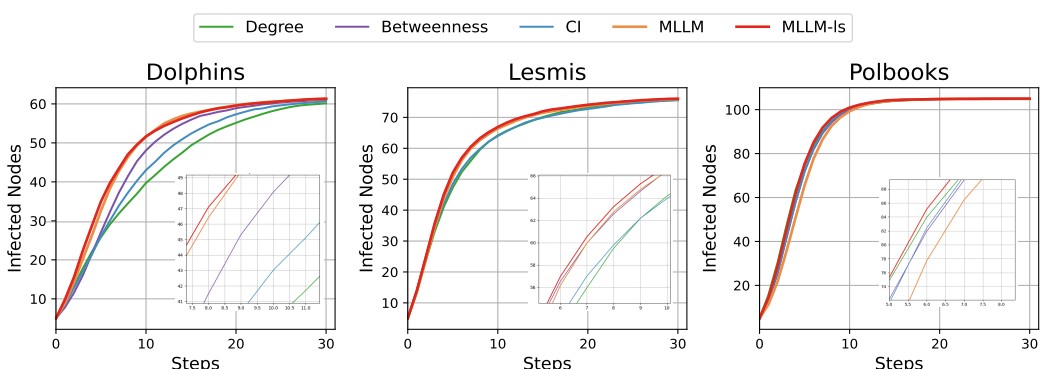

Figure 15: The influence maximization performance across different methods and large-scale networks. The spreading model is the SI model.

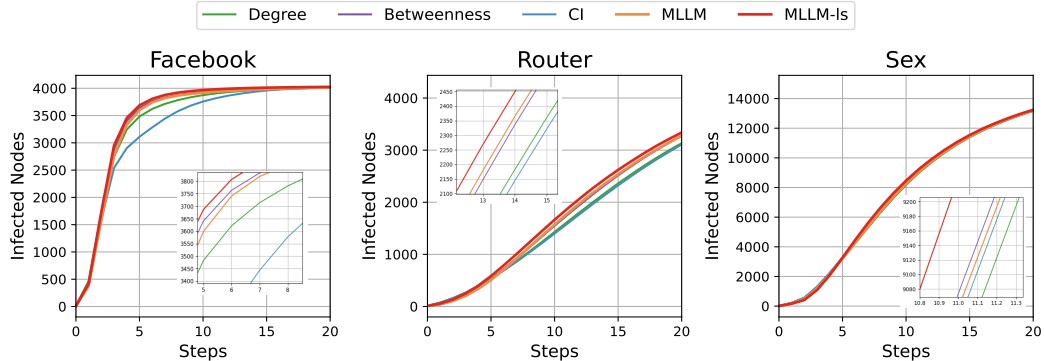

Figure 16: The influence maximization performance across different methods and large-scale networks. The spreading model is the SI model.

Table 10: The area under the curve (AUC) of different methods in influence maximization based on the SI model.

| Network | Dolphins | Lesmis | Polbooks | Facebook | Router | Sex |
|---|---|---|---|---|---|---|
| Degree | 1.31E3 | 1.87E3 | 2.74E3 | 1.08E5 | 6.61E4 | 2.87E5 |
| Betweenness | 1.42E3 | 1.91E3 | 2.71E3 | 1.09E5 | 6.88E4 | 2.89E5 |
| CI | 1.36E3 | 1.87E3 | 2.72E3 | 1.06E5 | 6.55E4 | 2.88E5 |
| MLLM | 1.48E3 | 1.91E3 | 2.68E3 | 1.09E5 | 6.90E4 | 2.88E5 |
| MLLM-ls | **1.49E3** | **1.92E3** | **2.74E3** | **1.10E5** | **7.08E4** | **2.90E5** |

The step-wise infected nodes for various methods are illustrated in Figures 17 and 18. The MLLM-based approach initially lags behind other centrality methods but ultimately achieves the highest spreading speed after several steps, indicating that MLLMs prioritize a widespread selection.

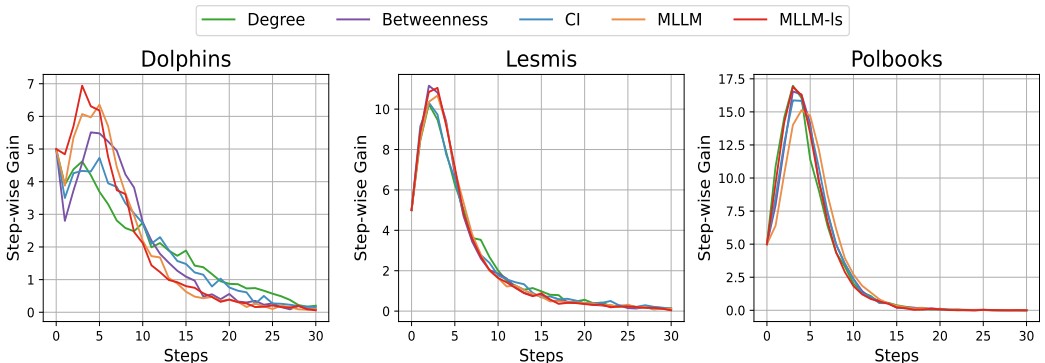

Figure 17: The step-wise infected nodes across different methods on small-scale networks. The spreading model is the SI model.

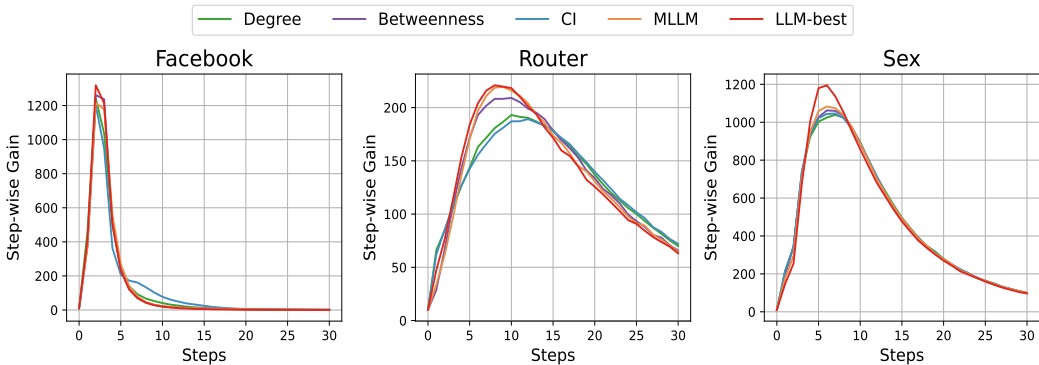

Figure 18: The step-wise infected nodes across different methods on large-scale networks. The spreading model is the SI model.

A.10 STATISTICAL RESULT OF THE DIFFERENT AGENTS

The results from the Tables 11 and 12 reveal that improved agents (denoted as "ls") consistently outperform normal agents across all networks and models in terms of influence spread. For instance, in the Dolphins network under the IC model, the average influence spread for Agent 1(ls) is 10.45 compared to 9.32 for the normal agent, with similar trends observed in the LT model. This improvement is not only reflected in higher average values but also in lower standard deviations, indicating more stable performance for the improved agents. Across all networks, the LT model generally achieves a higher influence spread than the IC model, although it tends to have greater variability.

In the Polbooks network, the difference between normal and improved agents is most pronounced, especially in the LT model, where Agent 1(ls) achieves an average influence spread of 51.64, compared to 33.90 for the normal agent. This suggests that the improved agents are particularly effective in

networks with more complex structures like Polbooks. Overall, the results demonstrate that improved agents offer substantial benefits in both influence spread and stability, making them a more reliable choice.

Table 11: Statistical performance of agents on small-scale networks (IC and LT Models)

| Network | Agent | IC Model (Avg ± Std) | IC Max | LT Model (Avg ± Std) | LT Max |
|---|---|---|---|---|---|
| **Dolphins** | Agent 1 | **9.32 ± 0.66** | **10.41** | **19.25 ± 2.19** | 23.73 |
| | Agent 2 | 9.22 ± 0.47 | 10.33 | 19.13 ± 2.03 | **23.80** |
| | Agent 3 | 9.12 ± 0.58 | 9.87 | 17.78 ± 1.72 | 20.25 |
| **Lesmis** | Agent 1 | 19.21 ± 0.91 | 20.46 | 42.65 ± 5.95 | 49.05 |
| | Agent 2 | **19.57 ± 0.71** | 20.24 | 44.39 ± 4.40 | 47.42 |
| | Agent 3 | 19.04 ± 0.74 | 20.10 | **44.74 ± 2.33** | 47.35 |
| **Polbooks** | Agent 1 | 22.29 ± 0.83 | 24.43 | 33.90 ± 1.78 | 37.29 |
| | Agent 2 | 20.71 ± 0.77 | 21.84 | 29.34 ± 2.00 | 32.47 |
| | Agent 3 | **22.53 ± 1.28** | **25.37** | **34.48 ± 2.69** | **39.45** |

Table 12: Statistical performance of agents on small-scale networks (IC and LT Models).

| Network | Agent | IC Model (Avg ± Std) | IC Max | LT Model (Avg ± Std) | LT Max |
|---|---|---|---|---|---|
| **Dolphins** | Agent 1(ls) | **10.45 ± 0.34** | **11.23** | **24.26 ± 1.33** | **26.79** |
| | Agent 2(ls) | 10.25 ± 0.40 | 11.07 | 23.19 ± 1.54 | 25.47 |
| | Agent 3(ls) | 10.13 ± 0.42 | 10.64 | 21.98 ± 0.96 | 23.68 |
| **Lesmis** | Agent 1(ls) | 20.12 ± 0.52 | 20.73 | 48.88 ± 2.62 | 52.11 |
| | Agent 2(ls) | **20.31 ± 0.33** | 20.72 | **50.05 ± 3.16** | 52.20 |
| | Agent 3(ls) | 19.93 ± 0.70 | **20.76** | 49.68 ± 2.42 | **52.25** |
| **Polbooks** | Agent 1(ls) | 28.77 ± 0.54 | 29.62 | 51.64 ± 2.62 | 54.39 |
| | Agent 2(ls) | 28.27 ± 0.60 | 29.09 | 50.21 ± 1.19 | 52.66 |
| | Agent 3(ls) | **28.95 ± 0.55** | **29.46** | **52.04 ± 1.28** | **54.31** |

As observed from Tables 13 and 14, MLLMs exhibit the similar trend to small-scale networks. specifically, local search greatly improves performance in both models, but especially in the LT model, where influence spread is larger and more stable (as seen in the Sex and Facebook networks). It enhances agents' ability to spread influence, reduces variability, and maximizes performance, particularly for Agent 2(ls) and Agent 3(ls). On the other hand, different agents are better suited to specific network structures and models, and there is no one-size-fits-all agent that performs optimally across all tests.

Table 13: Statistical performance of agents on large-scale networks (IC and LT Models).

| Network | Agent | IC Model (Avg ± Std) | IC Max | LT Model (Avg ± Std) | LT Max |
|---|---|---|---|---|---|
| **Facebook** | Agent 1 | 3046.26 ± 4.74 | **3055.81** | **959.30 ± 243.18** | **1231.05** |
| | Agent 2 | **3054.41 ± 3.49** | 3055.80 | 830.50 ± 192.82 | 1147.91 |
| | Agent 3 | 2906.04 ± 0.67 | 2907.06 | 878.75 ± 112.02 | 1030.16 |
| | Agent 4 | 2932.99 ± 8.38 | 2948.66 | 772.39 ± 189.69 | 1163.17 |
| **Router** | Agent 1 | 28.11 ± 1.90 | 31.07 | 30.46 ± 1.07 | 31.61 |
| | Agent 2 | **29.01 ± 2.06** | **31.65** | **31.08 ± 0.65** | **32.48** |
| | Agent 3 | 27.24 ± 1.51 | 29.77 | 21.10 ± 0.87 | 21.99 |
| | Agent 4 | 28.43 ± 1.70 | 30.71 | 26.41 ± 3.33 | 29.95 |
| **Sex** | Agent 1 | **2091.57 ± 28.36** | 2113.97 | 169.99 ± 113.27 | 348.01 |
| | Agent 2 | 2061.72 ± 72.64 | 2119.03 | 200.69 ± 139.18 | 386.32 |
| | Agent 3 | 2079.48 ± 49.14 | 2119.47 | 146.13 ± 38.63 | 198.49 |
| | Agent 4 | 2068.18 ± 65.39 | **2120.55** | **232.76 ± 144.50** | **429.11** |

Table 14: Statistical performance of agents with local search on large-scale networks (IC and LT Models)

| Network | Agent | IC Model (Avg ± Std) | IC Max | LT Model (Avg ± Std) | LT Max |
|---------|-------|----------------------|--------|----------------------|--------|
| Facebook | Agent 1(ls) | 3046.64 ± 4.60 | 3055.86 | **1291.22 ± 176.43** | 1416.00 |
| | Agent 2(ls) | **3055.04 ± 3.49** | **3057.20** | 1288.87 ± 99.46 | **1441.37** |
| | Agent 3(ls) | 2922.67 ± 11.87 | 2938.59 | 1110.16 ± 56.90 | 1203.64 |
| | Agent 4(ls) | 2935.82 ± 9.54 | 2958.40 | 1081.52 ± 150.70 | 1366.22 |
| Router | Agent 1(ls) | 32.91 ± 2.22 | 36.57 | 33.05 ± 2.71 | 38.69 |
| | Agent 2(ls) | **34.73 ± 3.16** | **38.94** | **33.42 ± 3.31** | **42.52** |
| | Agent 3(ls) | 31.23 ± 2.07 | 34.60 | 22.65 ± 1.76 | 25.90 |
| | Agent 4(ls) | 32.39 ± 3.28 | 36.84 | 30.03 ± 6.07 | 40.55 |
| Sex | Agent 1(ls) | 2108.86 ± 13.03 | 2124.44 | 199.41 ± 116.23 | 411.14 |
| | Agent 2(ls) | 2101.70 ± 36.34 | 2122.25 | 233.92 ± 139.10 | 423.67 |
| | Agent 3(ls) | 2113.95 ± 11.53 | 2121.01 | 230.68 ± 114.66 | **484.26** |
| | Agent 4(ls) | **2114.35 ± 7.39** | **2129.76** | **264.46 ± 141.17** | 454.06 |

### A.11 RESULTS OF SEED SELECTION

Figure 19 presents the input to MLLMs when optimizing on large-scale networks. As seen, it is difficult to display all the label in a limited canvas at the stage. We also need to avoid the displayed label too close to recognize. A feasible way is increasing the canvas to accommodate more labels. Ultimately, we aim to connect MLLMs with visualization software, enabling MLLMs to perform micro-observations and retrieve labels for any nodes.

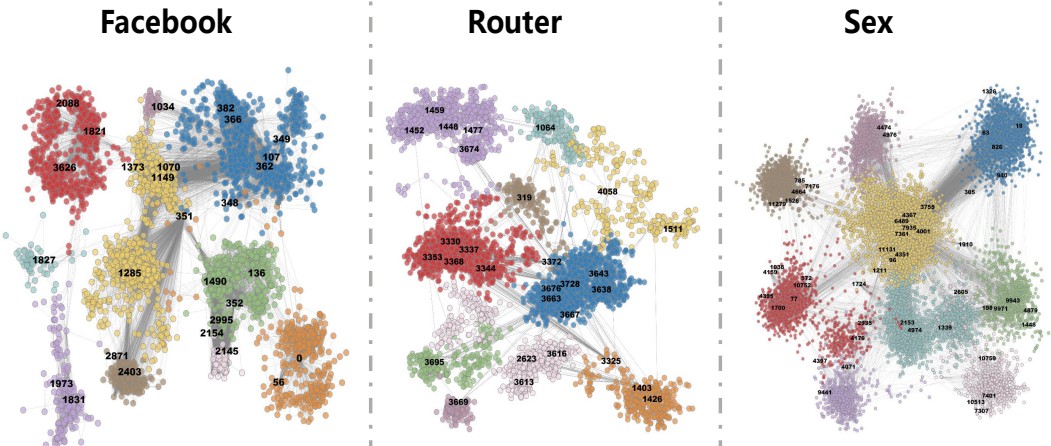

Figure 19: The original input images of three large-scale netwroks to the MLLM in partial-label case. Only a certain ratio of high-degree nodes of each network are displayed in the image where only the higher-degree one can be retained if there are two high-degree nodes that are too close to visually recognize their IDs.

Figures 11 and 21 presents the results of distribution of selected seeds by different agents on Facebook and Sex networks. Figures 12 and 23 illustrate the selected nodes, showing similarities to Router's results. The selection of all ten attempts on partial-label case is shown in Figures 24-35.

Figure 36 shows the input image to MLLMs in the full-label case. The displayed network will become dense as the network size increases. In Polbooks networks, some low-degree nodes appear high-degree due to numerous intersecting lines, making it difficult to distinguish them from actual high-degree nodes (see case (b) in Figure 14). Figures 37-45 illustrate the selected seeds in the full-label networks. The selected nodes of Agent 2 are more distributed, while those of Agent 3 are concentrated in specific areas, demonstrating MLLMs' spatial awareness of the graph structure.

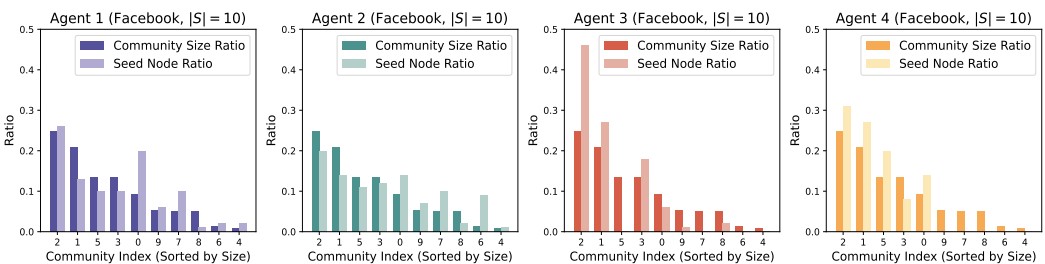

Figure 20: The distribution of ten selected seed nodes by different agents on the Facebook network. The bars refer to the community size ratio (darker bars) alongside the seed node ratio (lighter bars) for various community indices, which are sorted by size.

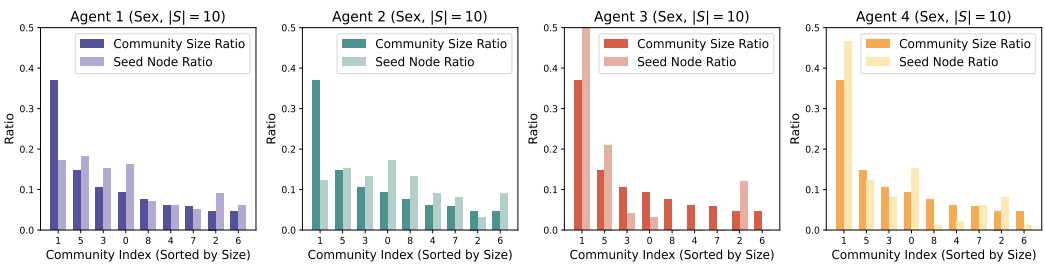

Figure 21: The distribution of ten selected seed nodes by different agents on the Sex network. The bars refer to the community size ratio (darker bars) alongside the seed node ratio (lighter bars) for various community indices, which are sorted by size.

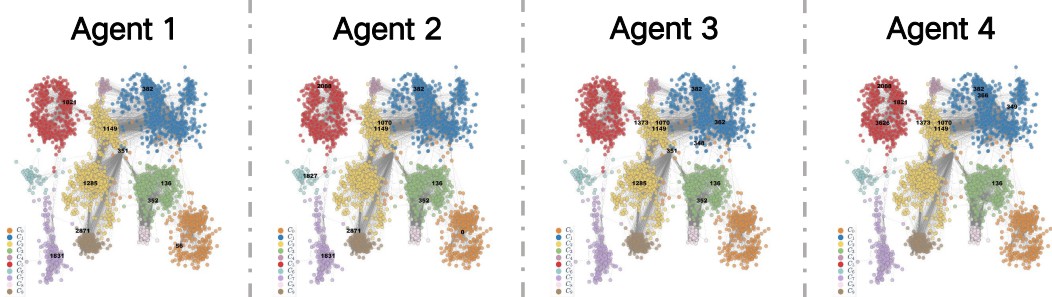

Figure 22: The selected nodes of different MLLM agents on the Facebook network. The seed size is ten.

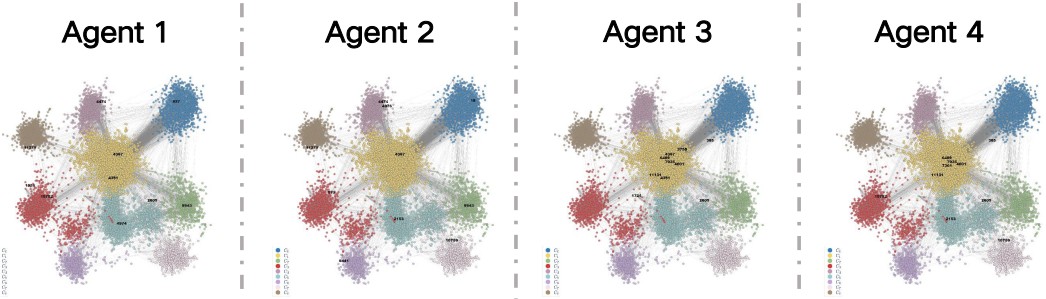

Figure 23: The selected nodes of different MLLM agents on the Sex network. The seed size is ten.

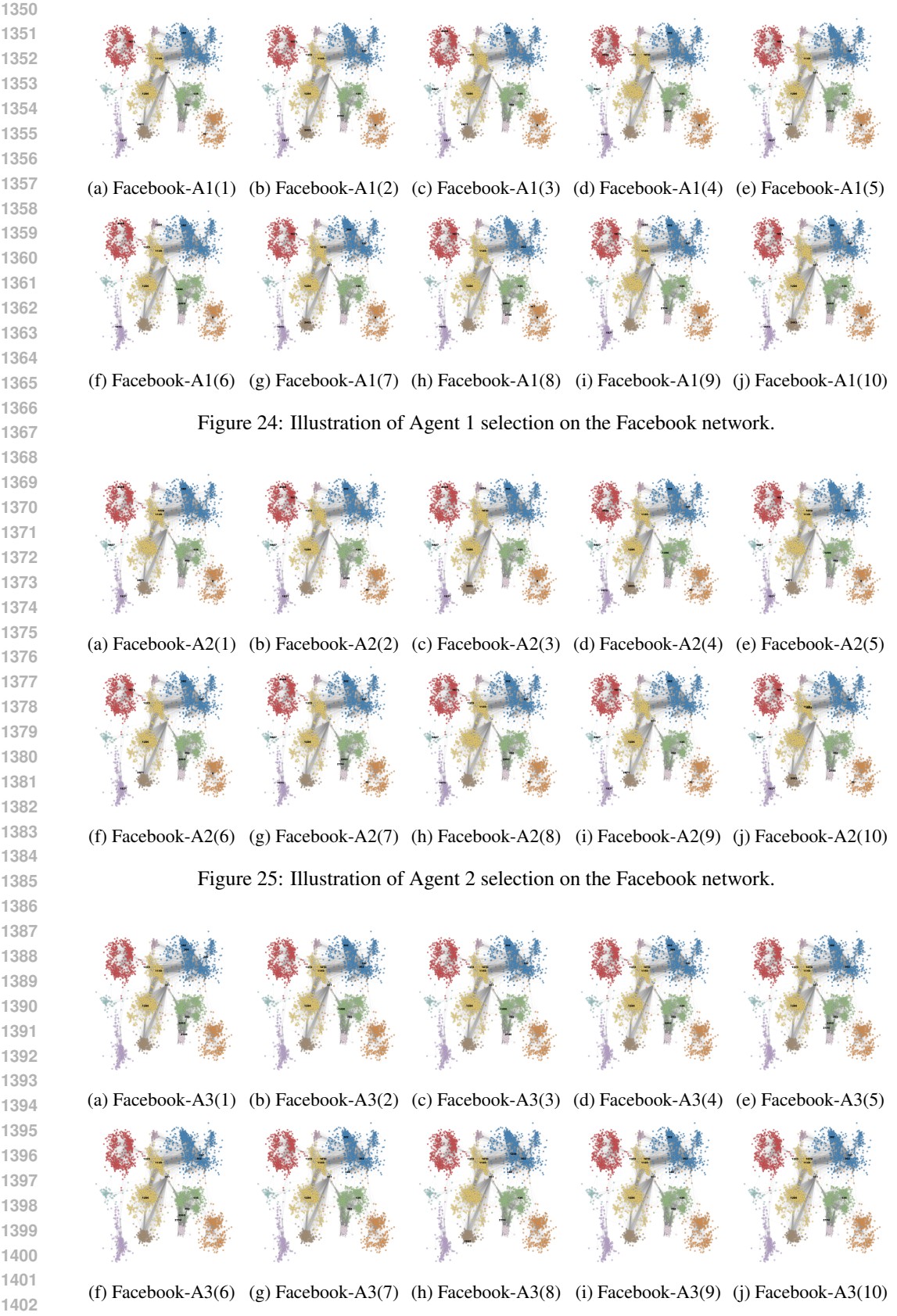

(a) Facebook-A1(1)   (b) Facebook-A1(2)   (c) Facebook-A1(3)   (d) Facebook-A1(4)   (e) Facebook-A1(5)

(f) Facebook-A1(6)   (g) Facebook-A1(7)   (h) Facebook-A1(8)   (i) Facebook-A1(9)   (j) Facebook-A1(10)

Figure 24: Illustration of Agent 1 selection on the Facebook network.

(a) Facebook-A2(1)   (b) Facebook-A2(2)   (c) Facebook-A2(3)   (d) Facebook-A2(4)   (e) Facebook-A2(5)

(f) Facebook-A2(6)   (g) Facebook-A2(7)   (h) Facebook-A2(8)   (i) Facebook-A2(9)   (j) Facebook-A2(10)

Figure 25: Illustration of Agent 2 selection on the Facebook network.

(a) Facebook-A3(1)   (b) Facebook-A3(2)   (c) Facebook-A3(3)   (d) Facebook-A3(4)   (e) Facebook-A3(5)

(f) Facebook-A3(6)   (g) Facebook-A3(7)   (h) Facebook-A3(8)   (i) Facebook-A3(9)   (j) Facebook-A3(10)

Figure 26: Illustration of Agent 3 selection on the Facebook network.

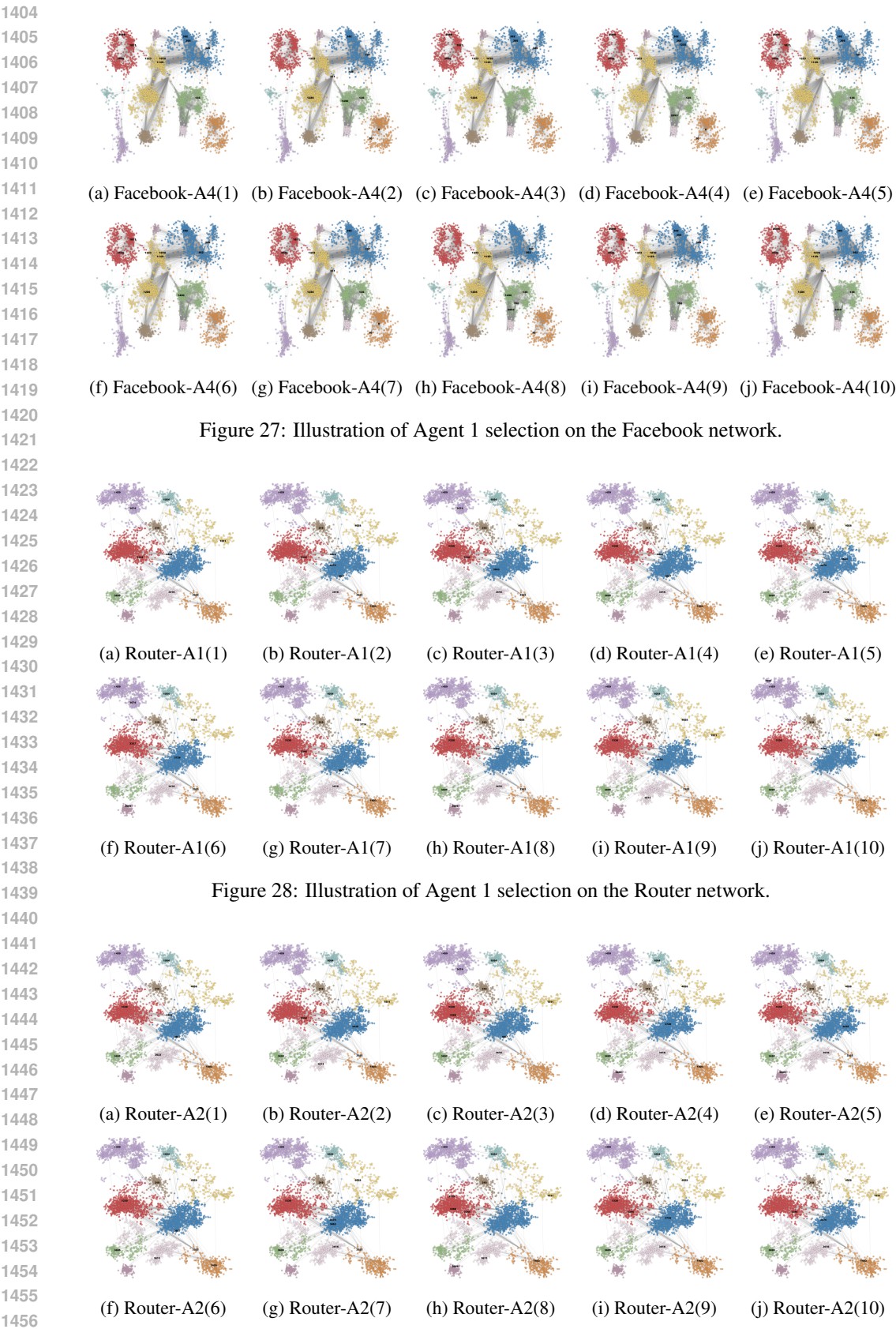

(a) Facebook-A4(1)  (b) Facebook-A4(2)  (c) Facebook-A4(3)  (d) Facebook-A4(4)  (e) Facebook-A4(5)

(f) Facebook-A4(6)  (g) Facebook-A4(7)  (h) Facebook-A4(8)  (i) Facebook-A4(9)  (j) Facebook-A4(10)

Figure 27: Illustration of Agent 1 selection on the Facebook network.

(a) Router-A1(1)  (b) Router-A1(2)  (c) Router-A1(3)  (d) Router-A1(4)  (e) Router-A1(5)

(f) Router-A1(6)  (g) Router-A1(7)  (h) Router-A1(8)  (i) Router-A1(9)  (j) Router-A1(10)

Figure 28: Illustration of Agent 1 selection on the Router network.

(a) Router-A2(1)  (b) Router-A2(2)  (c) Router-A2(3)  (d) Router-A2(4)  (e) Router-A2(5)

(f) Router-A2(6)  (g) Router-A2(7)  (h) Router-A2(8)  (i) Router-A2(9)  (j) Router-A2(10)

Figure 29: Illustration of Agent 2 selection on the Router network.

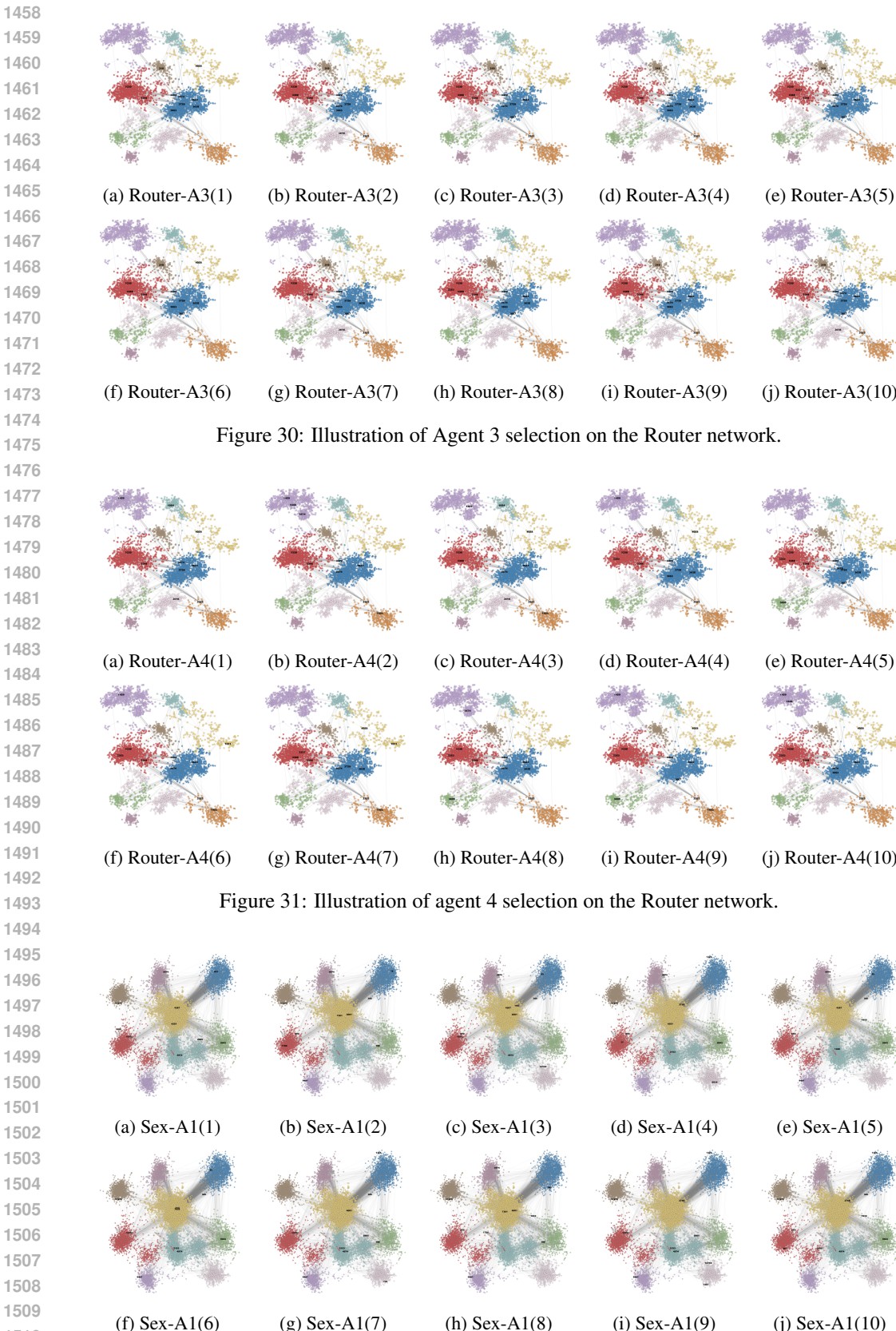

(a) Router-A3(1)  (b) Router-A3(2)  (c) Router-A3(3)  (d) Router-A3(4)  (e) Router-A3(5)

(f) Router-A3(6)  (g) Router-A3(7)  (h) Router-A3(8)  (i) Router-A3(9)  (j) Router-A3(10)

Figure 30: Illustration of Agent 3 selection on the Router network.

(a) Router-A4(1)  (b) Router-A4(2)  (c) Router-A4(3)  (d) Router-A4(4)  (e) Router-A4(5)

(f) Router-A4(6)  (g) Router-A4(7)  (h) Router-A4(8)  (i) Router-A4(9)  (j) Router-A4(10)

Figure 31: Illustration of agent 4 selection on the Router network.

(a) Sex-A1(1)  (b) Sex-A1(2)  (c) Sex-A1(3)  (d) Sex-A1(4)  (e) Sex-A1(5)

(f) Sex-A1(6)  (g) Sex-A1(7)  (h) Sex-A1(8)  (i) Sex-A1(9)  (j) Sex-A1(10)

Figure 32: Illustration of Agent 1 selection on the Sex network.

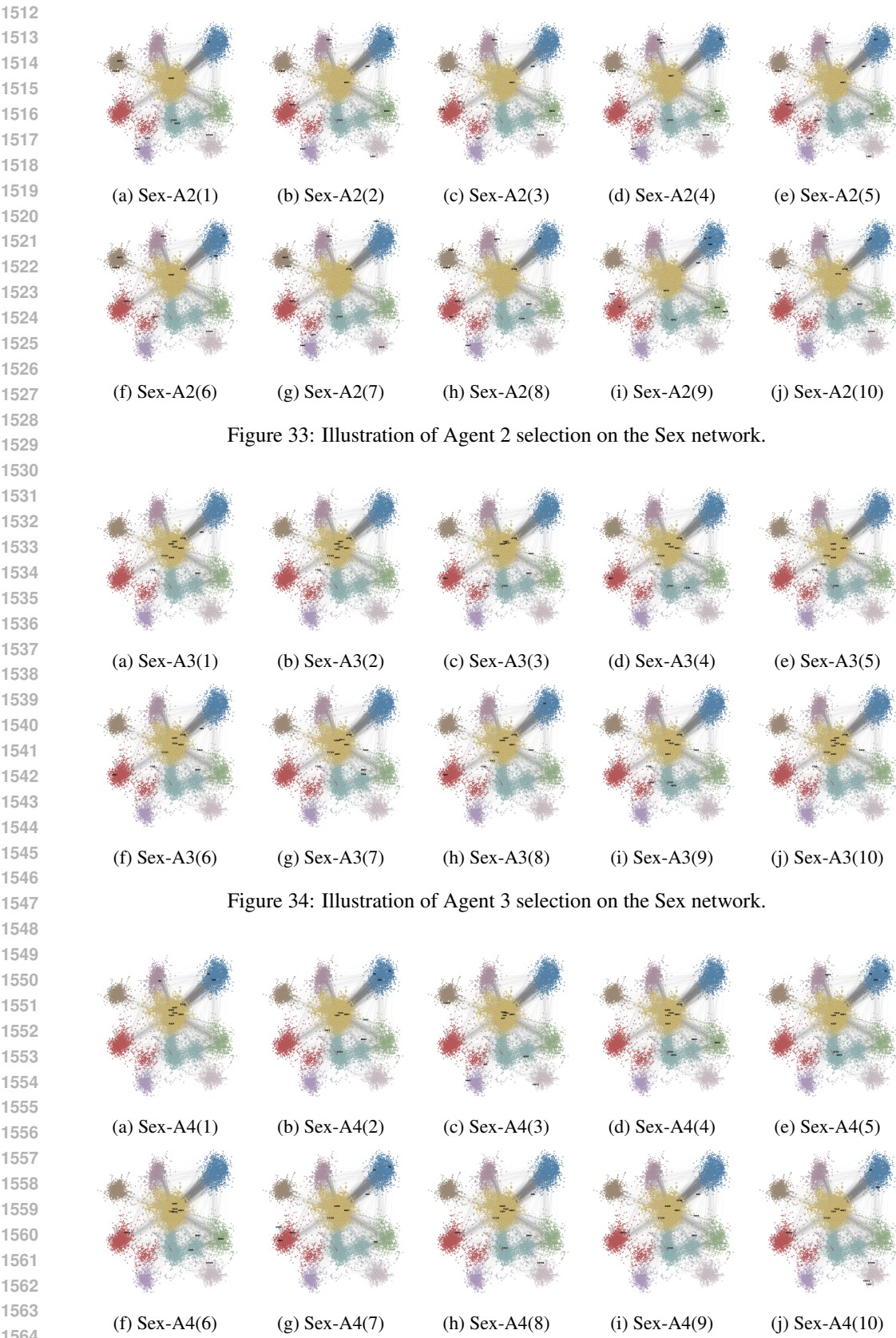

(a) Sex-A2(1)  (b) Sex-A2(2)  (c) Sex-A2(3)  (d) Sex-A2(4)  (e) Sex-A2(5)

(f) Sex-A2(6)  (g) Sex-A2(7)  (h) Sex-A2(8)  (i) Sex-A2(9)  (j) Sex-A2(10)

Figure 33: Illustration of Agent 2 selection on the Sex network.

(a) Sex-A3(1)  (b) Sex-A3(2)  (c) Sex-A3(3)  (d) Sex-A3(4)  (e) Sex-A3(5)

(f) Sex-A3(6)  (g) Sex-A3(7)  (h) Sex-A3(8)  (i) Sex-A3(9)  (j) Sex-A3(10)

Figure 34: Illustration of Agent 3 selection on the Sex network.

(a) Sex-A4(1)  (b) Sex-A4(2)  (c) Sex-A4(3)  (d) Sex-A4(4)  (e) Sex-A4(5)

(f) Sex-A4(6)  (g) Sex-A4(7)  (h) Sex-A4(8)  (i) Sex-A4(9)  (j) Sex-A4(10)

Figure 35: Illustration of Agent 4 selection on the Sex network.

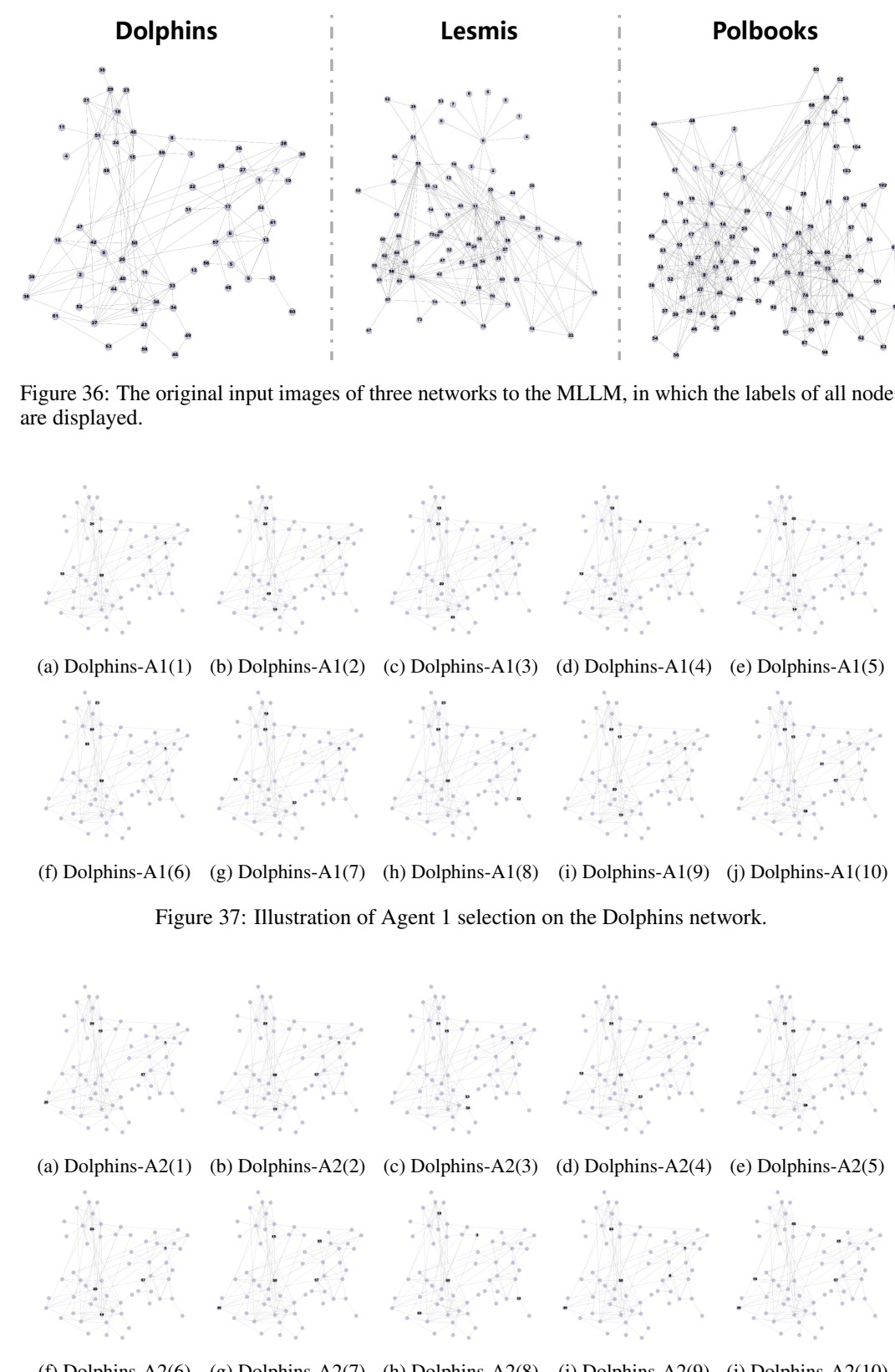

Figure 36: The original input images of three networks to the MLLM, in which the labels of all nodes are displayed.

(a) Dolphins-A1(1)  (b) Dolphins-A1(2)  (c) Dolphins-A1(3)  (d) Dolphins-A1(4)  (e) Dolphins-A1(5)

(f) Dolphins-A1(6)  (g) Dolphins-A1(7)  (h) Dolphins-A1(8)  (i) Dolphins-A1(9)  (j) Dolphins-A1(10)

Figure 37: Illustration of Agent 1 selection on the Dolphins network.

(a) Dolphins-A2(1)  (b) Dolphins-A2(2)  (c) Dolphins-A2(3)  (d) Dolphins-A2(4)  (e) Dolphins-A2(5)

(f) Dolphins-A2(6)  (g) Dolphins-A2(7)  (h) Dolphins-A2(8)  (i) Dolphins-A2(9)  (j) Dolphins-A2(10)

Figure 38: Illustration of Agent 2 selection on the Dolphins network.

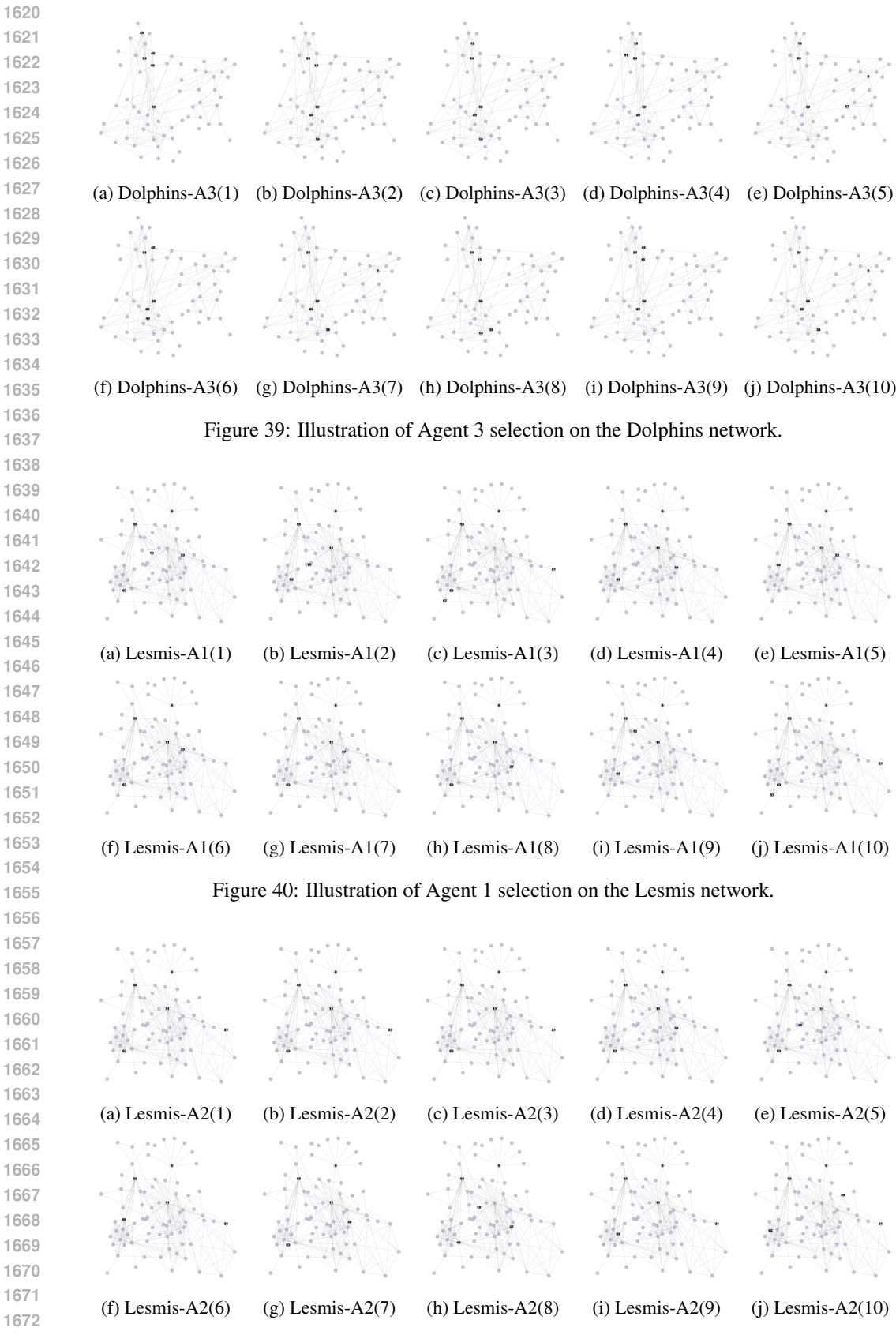

(a) Dolphins-A3(1)  (b) Dolphins-A3(2)  (c) Dolphins-A3(3)  (d) Dolphins-A3(4)  (e) Dolphins-A3(5)

(f) Dolphins-A3(6)  (g) Dolphins-A3(7)  (h) Dolphins-A3(8)  (i) Dolphins-A3(9)  (j) Dolphins-A3(10)

Figure 39: Illustration of Agent 3 selection on the Dolphins network.

(a) Lesmis-A1(1)  (b) Lesmis-A1(2)  (c) Lesmis-A1(3)  (d) Lesmis-A1(4)  (e) Lesmis-A1(5)

(f) Lesmis-A1(6)  (g) Lesmis-A1(7)  (h) Lesmis-A1(8)  (i) Lesmis-A1(9)  (j) Lesmis-A1(10)

Figure 40: Illustration of Agent 1 selection on the Lesmis network.

(a) Lesmis-A2(1)  (b) Lesmis-A2(2)  (c) Lesmis-A2(3)  (d) Lesmis-A2(4)  (e) Lesmis-A2(5)

(f) Lesmis-A2(6)  (g) Lesmis-A2(7)  (h) Lesmis-A2(8)  (i) Lesmis-A2(9)  (j) Lesmis-A2(10)

Figure 41: Illustration of Agent 2 selection on the Lesmis network.

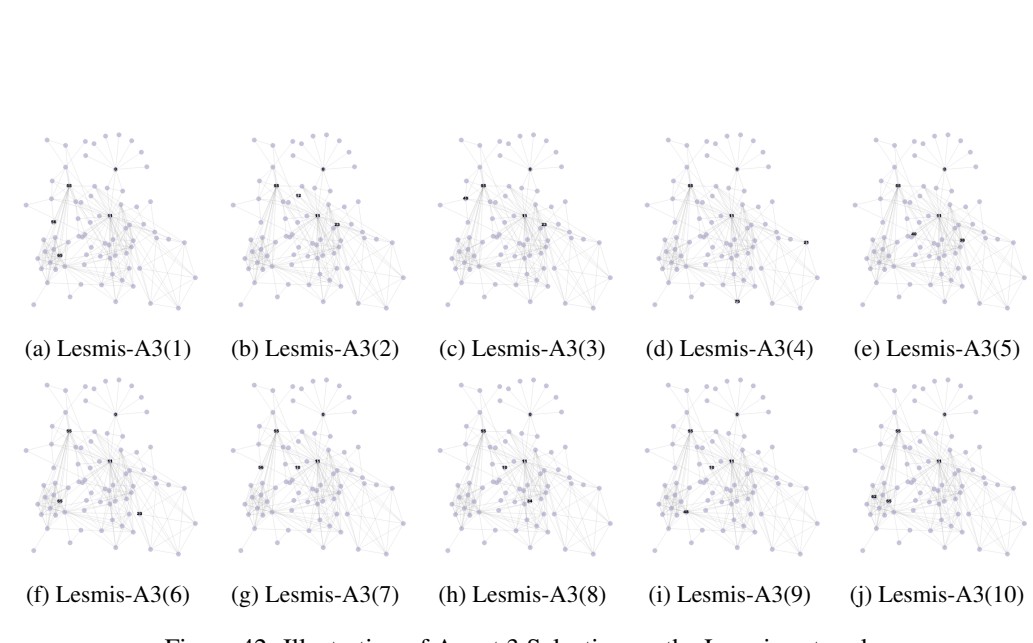

(a) Lesmis-A3(1)    (b) Lesmis-A3(2)    (c) Lesmis-A3(3)    (d) Lesmis-A3(4)    (e) Lesmis-A3(5)

(f) Lesmis-A3(6)    (g) Lesmis-A3(7)    (h) Lesmis-A3(8)    (i) Lesmis-A3(9)    (j) Lesmis-A3(10)

Figure 42: Illustration of Agent 3 Selection on the Lesmis network.

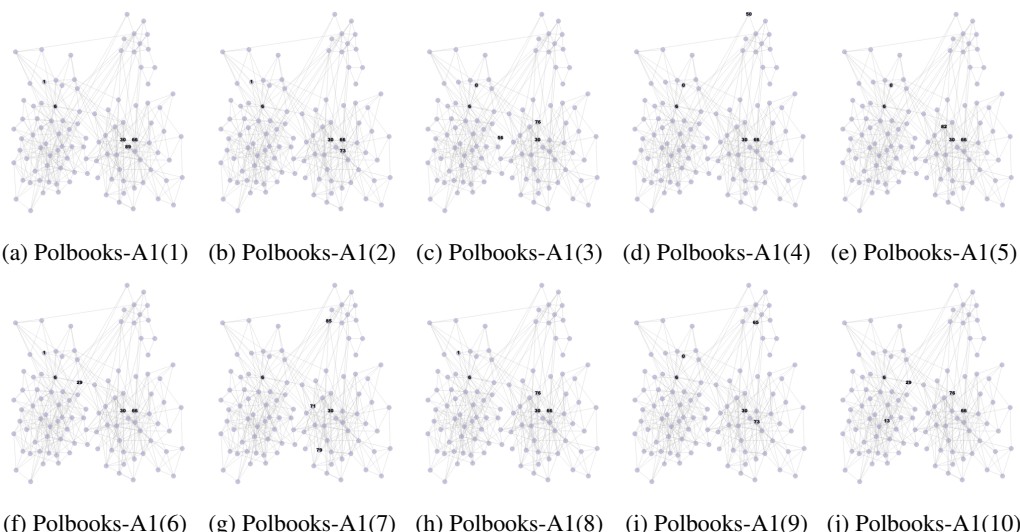

(a) Polbooks-A1(1)    (b) Polbooks-A1(2)    (c) Polbooks-A1(3)    (d) Polbooks-A1(4)    (e) Polbooks-A1(5)

(f) Polbooks-A1(6)    (g) Polbooks-A1(7)    (h) Polbooks-A1(8)    (i) Polbooks-A1(9)    (j) Polbooks-A1(10)

Figure 43: Illustration of Agent 1 selection on the Polbooks network.

(a) Polbooks-A2(1)  (b) Polbooks-A2(2)  (c) Polbooks-A2(3)  (d) Polbooks-A2(4)  (e) Polbooks-A2(5)

(f) Polbooks-A2(6)  (g) Polbooks-A2(7)  (h) Polbooks-A2(8)  (i) Polbooks-A2(9)  (j) Polbooks-A2(10)

Figure 44: Illustration of Agent 2 selection on the Polbooks network.

(a) Polbooks-A3(1)  (b) Polbooks-A3(2)  (c) Polbooks-A3(3)  (d) Polbooks-A3(4)  (e) Polbooks-A3(5)

(f) Polbooks-A3(6)  (g) Polbooks-A3(7)  (h) Polbooks-A3(8)  (i) Polbooks-A3(9)  (j) Polbooks-A3(10)

Figure 45: Illustration of Agent 3 selection on the Polbooks network.

