# OpenReview forum: "Spatial Reasoning with MLLMs: A New Path to Graph-Structured Optimization"
_ICLR.cc/2025/Conference — ICLR 2025 Conference Withdrawn Submission_

### Official Review · Reviewer_GoHR · 2024-11-02

**Soundness:** 2
**Presentation:** 1
**Contribution:** 1
**Rating:** 3
**Confidence:** 5

**Summary:**

This paper demonstrates the unique advantages and promising future of representing graph structures as images and using MLLM for reasoning on graph combinational and basic reasoning problems. Specifically, it shows the improvements of MLLM over some simple traditional methods in tasks like influence maximization and network dismantling, as well as its superiority over LLM in six fundamental graph problems. Additionally, the paper analyzes several detailed factors that affect performance when using MLLM with visual representations of graph structures for graph reasoning, including color, layout, the random generation form of graph structures, and scale.

**Strengths:**

1. The paper focuses on an emerging topic: representing graphs as images and using MLLM for reasoning. It demonstrates and analyzes the advantages of MLLM over traditional methods in various tasks.

2. Experiments on two combinatorial graph optimization tasks, influence maximization and network dismantling, serve as a supplementary extension to previously explored graph reasoning tasks. These experiments further illustrate the potential of the emerging field of visual-language graph reasoning, providing valuable insights.

**Weaknesses:**

1. Missing Key References and Overstated Contributions: The paper lacks critical references and may exaggerate its contributions.
For instance, GITA [1] introduced a framework that systematically converts graphs to images and feeds them into MLLMs for seven fundamental graph reasoning tasks and two downstream tasks. It provides detailed comparisons with LLMs and GNNs to showcase the advantages of using MLLMs with image representations of graphs with its visual intelligence. Similarly, VisionGraph [2] explores leveraging large multimodal models for graph theory problems in a visual context, establishing a toolchain for eight complex graph problem tasks.
Therefore, the claimed core contribution of the paper, "introducing an original and novel approach by feeding graphs as images into MLLMs," has already been addressed in previous work. Furthermore, a significant portion of the findings in the paper overlap with those in GITA [1] and VisionGraph[2]. For example, the intuitive perception and understanding of graph structures by MLLMs; the zero-shot graph reasoning capability of MLLMs; MLLMs being better at tasks requiring global and visual perception; the significant impact of layout on performance in the image representation of graphs; MLLMs outperforming LLMs in graph-related tasks; and the challenges posed by the increasing complexity of visual representations of graphs (including the number of nodes and edges) as the graph scale increases, are also analyzed and mentioned in GITA [1] or VisionGraph [2].
Therefore, the paper seems to exaggerate its contributions and lacks distinct novelty and constructive findings. Future versions should clearly differentiate and highlight their unique contributions, perhaps by focusing on specific scenarios in combinatorial optimization problems.

2. Poor Writing Structure: The structure of the paper appears disorganized, with the introduction followed immediately by the experimental section. Discussions of related work and descriptions of the methodology are embedded within the introduction and experimental sections. This makes the paper resemble a brief experimental report rather than a polished academic paper. Additionally, some concepts and terms such as BA, ER, and WS are not adequately introduced or explained in the main text, making it difficult for readers to follow their divergences and properties.

3. Lack of Methodological and Experimental Details: The paper lacks crucial details about the methodology and experiments, such as how graph structures are converted into images and which specific MLLM and LLM are used in the experiments. These should be clearly explained in the main text. Furthermore, the model selection in the experimental setup is too narrow, and the baselines are insufficient, raising questions about the generality and robustness of the conclusions.

[1] GITA: Graph to Visual and Textual Integration for Vision-Language Graph Reasoning. https://arxiv.org/abs/2402.02130. NeurIPS 2024.

[2] VisionGraph: Leveraging Large Multimodal Models for Graph Theory Problems in Visual Context. https://arxiv.org/abs/2405.04950. ICML 2024.

**Questions:**

In addition to the concerns mentioned in the weaknesses section, I have the following questions:

1. Although the paper classifies graph scales as small-scale and large-scale, in all the experiments, the so-called "large-scale" graphs actually only consist of a few dozen nodes. This scale is still tiny. As far as I know, in the field of graph theory, large-scale more often refers to graphs with millions of nodes or even more. In such cases, as mentioned in GITA and in section 3.1 of this paper, too many nodes and edges crammed onto a canvas can turn it into an indistinguishable black smudge. Therefore, the claim in the abstract and section four that MLLMs are more capable of handling large-scale graphs does not seem valid.

2. Comparing multiple models of MLLMs with different specialized GNNs and various models of LLMs would be beneficial in demonstrating the performance upper and lower bounds of MLLMs, and could lead to deeper analysis and insights.

3. For heterogeneous graphs with node and edge types, weighted graphs with node and edge attributes (which may not only be numerical but also vector representations), and reasoning problems on graph structures exceeding thousands or tens of thousands of nodes, can MLLMs still be able to provide solutions for combinatorial problems as a foundational model, or limited to simple and tiny homogeneous graphs?

---

### Official Review · Reviewer_bS3h · 2024-11-03

**Soundness:** 2
**Presentation:** 2
**Contribution:** 2
**Rating:** 3
**Confidence:** 5

**Summary:**

The paper claims to introduce a method employing Multimodal Large Language Models (MLLMs) for addressing graph-structured problems by inputting graph representations as images. The study asserts that MLLMs demonstrate an ability to perform tasks such as network dismantling and influence maximization without complex tuning. The paper also suggests that the scalability of MLLMs could be advantageous for large network analysis. Also, it notes that the effectiveness of MLLMs is potentially limited by the current state of visualization tools, implying that improvements in this area could influence their performance.

**Strengths:**

1.Research Potential: The chosen research direction holds potential, especially in exploring the application of MLLMs in graph-structured optimization.
2.Visual Presentation: The figures and tables are presented clearly and aesthetically, which aids readability and enhances the presentation quality of the experimental results.

**Weaknesses:**

1.Experimental Setup Issues: The experiments in “4. MLLM on Basic Graph-Related Tasks" have significant flaws. The specific MLLM and LLM models used are not even explicitly mentioned. Even if model types were specified, inconsistencies in model selection and prompt settings prevent a fair assessment of MLLM performance, as variables are not properly controlled. This lack of experimental rigor weakens the claim that MLLMs outperform LMMs.
2.Simplistic Conclusions with Limited Analysis: Some conclusions are overly idealistic and lack detailed analysis. For example, the statement that “regardless of how large or complex the network is, the input remains a fixed-size image, allowing the MLLM to interpret and process it efficiently” is an oversimplification. In reality, MLLMs often struggle to accurately recognize nodes and edges, leading to misinterpretation of even basic structural information. Based on this, high-level insights tend to be guesses rather than grounded outputs. So the results provided by MLLMs often lack interpretability and practical feasibility. And based on this, the claim that “MLLM is able to process global relationships in graphs efficiently” needs more convincing experiments.
3.Lack of Novelty and Credible Proof: The paper attempts to showcase a well-known conclusion that the potential of MLLMs in graph-structured optimization. However, the experimental approach is not rigorous, and the absence of in-depth case studies undermines the persuasiveness of the arguments. Overall, the paper resembles an incomplete experimental report, lacking the necessary theoretical explanations, methodological robustness, and compelling case studies to support its conclusions.

**Questions:**

Please refer to the weakness.

---

### Official Review · Reviewer_Vq69 · 2024-11-03

**Soundness:** 3
**Presentation:** 3
**Contribution:** 3
**Rating:** 6
**Confidence:** 3

**Summary:**

This paper introduces a novel approach to solving graph-structured problems by leveraging multimodal large language models (MLLMs) to visually represent graphs as images, facilitating spatial reasoning in tasks like network dismantling and influence maximization. The authors explore the potential of MLLMs to capture graph structures intuitively without extensive training or computationally demanding optimizations. Experimental results demonstrate that MLLMs achieve promising results, often outperforming traditional graph neural networks (GNNs) and heuristic-based methods. This work marks a step towards integrating MLLMs into graph-based problem-solving, leveraging visual cues to enhance computational efficiency and effectiveness.

**Strengths:**

- The paper includes extensive experiments across multiple graph-related tasks, demonstrating MLLMs' superior performance over baseline methods in network dismantling and influence maximization.

- The methodology is well-structured, and the visual examples provided aid in understanding how MLLMs process graph data, making the approach accessible and informative for the reader.

- This approach offers a promising alternative to traditional graph processing techniques, highlighting the potential for MLLMs to simplify complex graph-based tasks, which could have wide-ranging implications across fields requiring large-scale graph analysis.

**Weaknesses:**

- The paper relies on specific layout methods (e.g., force-directed layouts) to structure graphs in ways that facilitate MLLM processing. However, this dependency may limit generalization to more complex graphs with less intuitive visual layouts, potentially affecting the model's versatility.

- The current approach lacks interactive visualization tools, such as zooming capabilities, that could enhance MLLMs’ performance in dense networks. This limitation affects MLLMs' ability to accurately analyze highly detailed or node-dense regions.

- While the paper explores common graph-related tasks, the evaluation is limited to specific problem types and graph structures. Further testing on diverse graph types, such as dynamic or weighted graphs, would provide a more comprehensive assessment of MLLMs' effectiveness.

**Questions:**

- Can the proposed MLLM approach handle dynamic or evolving graphs, where node connections may change over time?

- How would the model perform on weighted graphs where edges carry additional information? Are there modifications that could integrate edge weights into the visual representation?

- Could an interactive visualization framework improve the scalability of the model for more complex or larger graphs?

---

### Official Review · Reviewer_idxA · 2024-11-03

**Soundness:** 2
**Presentation:** 2
**Contribution:** 2
**Rating:** 5
**Confidence:** 5

**Summary:**

This paper proposes to increase the performance of spatial reasoning on graph, which is achieved by transforming the graphs into images, and then feeding the images into the multimodal LLMs.

**Strengths:**

The writing is good.

**Weaknesses:**

There are several confusions should be addressed:

1) Given the importance of trasnformation from graph to image, the transforming manner is also important. Therefore, the transforming/imaging manner should also be discussed, in case there is any prior knowledge from human are invloved, which may leads to unfair comparison.

2) There is typos in the caption of Fig(5), only three agents are presented.

3) For the graph visualization. How to deal with complex graphs, such as the hetergeneous ones? These complex graphs are more common in the real-world problems compared with the simple ones presented in the paper.

4) Fig(7) and (9) are confusing and lacking of explanation. What is the LS? What do the lines of degree, betweeness, and closeness mean? It is hard to understand why it shows the intelligence of the MLLM for structural understanding.

5) For the large-scale dataset, the color and chosed degrees for construting the image already involves prior knowledge from human, which could lead to unfair comparison with conventional methods.

**Questions:**

Please refer to the weaknesses.

---

### Official Review · Reviewer_LEBR · 2024-11-03

**Soundness:** 2
**Presentation:** 3
**Contribution:** 2
**Rating:** 3
**Confidence:** 4

**Summary:**

This manuscript explores using MLLM to understand the graph data. The authors feed the image of visualization of a graph into the MLLM and ask the model to solve several problems that involve the structure of the data. The results show that the MLLM, to some degree, can understand the visualization of graph data.

**Strengths:**

- It is an interesting attempt to assess how well the MLLM can interpret visualizations of graph data.

**Weaknesses:**

- The writing of the manuscript can be improved. For example, it is suggested to provide the full name of an abbreviated term as it's first used (HD, HCI)

- This manuscript, while addressing an intriguing problem, reads more like a report than a research paper. It examines the capability of MLLMs to interpret images containing structured data. I believe this manuscript might be suitable for another conference that focuses more on the application.

- The comparisons should contain some stronger heuristic methods to set the upper bound of the non-learning-based method.

- As the MLLM relies on the image of the graph, different visualization methods should be examined. Meanwhile, for larger networks and fine-grained tasks (like recommendations on social networks), visualization might be an inefficient or even impossible way to represent the graph.

**Questions:**

Please see the weaknesses part.

---

### Note · Authors · 2024-11-25

I have read and agree with the venue's withdrawal policy on behalf of myself and my co-authors.